# A versatile active learning workflow for optimization of genetic and metabolic networks

Amir Pandi ⬛ [1,9✉], Christoph Diehl[1,9], Ali Yazdizadeh Kharrazi[2], Scott A. Scholz ⬛ [1], Elizaveta Bobkova[1], Léon Faure[3], Maren Nattermann[1], David Adam[1], Nils Chapin[1], Yeganeh Foroughijabbari[1], Charles Moritz[1], Nicole Paczia[4], Niña Socorro Cortina[1,5], Jean-Loup Faulon[3,6,7] & Tobias J. Erb ⬛ [1,8✉]

Optimization of biological networks is often limited by wet lab labor and cost, and the lack of convenient computational tools. Here, we describe METIS, a versatile active machine learning workflow with a simple online interface for the data-driven optimization of biological targets with minimal experiments. We demonstrate our workflow for various applications, including cell-free transcription and translation, genetic circuits, and a 27-variable synthetic $CO_2$-fixation cycle (CETCH cycle), improving these systems between one and two orders of magnitude. For the CETCH cycle, we explore $10^{25}$ conditions with only 1,000 experiments to yield the most efficient $CO_2$-fixation cascade described to date. Beyond optimization, our workflow also quantifies the relative importance of individual factors to the performance of a system identifying unknown interactions and bottlenecks. Overall, our workflow opens the way for convenient optimization and prototyping of genetic and metabolic networks with customizable adjustments according to user experience, experimental setup, and laboratory facilities.

[1] Department of Biochemistry & Synthetic Metabolism, Max Planck Institute for Terrestrial Microbiology, Marburg, Germany. [2] DataChef, Amsterdam, The Netherlands. [3] Micalis Institute, INRAE, AgroParisTech, University of Paris-Saclay, Jouy-en-Josas, France. [4] Core Facility for Metabolomics and Small Molecule Mass Spectrometry, Max Planck Institute for Terrestrial Microbiology, Marburg, Germany. [5] LiVeritas Biosciences, Inc., 432N Canal St.; Ste. 20, South San Francisco, CA 94080, USA. [6] Genomique Metabolique, Genoscope, Institut Francois Jacob, CEA, CNRS, Univ Evry, University of Paris-Saclay, Evry, France. [7] Manchester Institute of Biotechnology, SYNBIOCHEM center, School of Chemistry, The University of Manchester, Manchester, UK. [8] SYNMIKRO Center of Synthetic Microbiology, Marburg, Germany. [9] These authors contributed equally: Amir Pandi, Christoph Diehl. ✉email: amir.pandi@mpi-marburg.mpg.de; toerb@mpi-marburg.mpg.de

The understanding and engineering of biological systems require practical and efficient experimental and computational approaches[1–5]. Machine learning algorithms hold a big promise for the study, design, and optimization of different biological systems[6–9], including genomics studies[10–12], protein, enzyme and metabolic engineering[4,13,14], prediction and optimization of CRISPR sequences and proteins[15–18], as well as complex genetic circuits design and optimization[19–21]. Yet, applying machine learning is limited by the need for informatics expertise and large user-labeled datasets, which are typically time-, labor- and cost-intensive.

Active learning, sometimes called optimal experimental design[22,23], is a type of machine learning that interactively suggests a next set of experiments after being trained on previous results[24]. This makes active learning valuable for wet-lab scientists, especially when dealing with a limited number of user-labeled data[25]. Active learning approaches reduce experimental time, labor and cost and have been used in cellular imaging[26], systems biology[27], biochemistry[28–30], and synthetic biology[31]. Despite these examples, a challenge in applying active learning methods for experimental biologists is the lack of customizable programs and workflows.

Here, aimed at democratization and standardization, we describe METIS (Machine-learning guided Experimental Trials for Improvement of Systems, named after the ancient goddess of wisdom and crafts Μῆτις, lit. "wise counsel"), a modular and versatile active machine learning workflow for data-driven optimization of a biological objective function (an output/target that depends on multiple factors) with minimal datasets. Note that, active learning for optimizing a system is also known as Bayesian optimization. We created METIS for experimentalists with no experience in programming, who can use the entire process of personalized active learning, experimental setup, data analysis and visualization without any advanced computational skills. METIS runs on Google Colab, a free online platform to write and execute Python codes developed for education, data science, and machine learning purposes[32]. The open platform does not need any installation/registration and local computational power and can be simply used via a personal copy of the respective notebook.

To establish the workflow, we first assessed the performance of different machine learning algorithms on a minimal training dataset and experimentally validated the best performing algorithm (XGBoost) by optimization of an in vitro cell-free transcription-translation (TXTL) system of Escherichia coli that is commonly used in cell-free synthetic biology for a variety of applications[33], including biosensor development[34], metabolic pathway prototyping[35], and gene circuit design[36]. We then developed the modular architecture of METIS for user-defined applications through the customization of different parameters and factors.

We showcase the versatility of METIS on various biological systems, starting with an in vitro gene circuit. Cell-free gene circuits have recently received attention (e.g., as biosensors), but are still limited in their applicability due to their poor performance[34,37]. Applying our workflow, we could improve the activity of a recently reported LacI-based multi-level controller[38] by two orders of magnitude, notably by identifying and overcoming a fundamental bottleneck (i.e., resource competition) in the design of the system. We further demonstrate ten-fold improved protein production from an optimized transcription & translation unit, demonstrating that our workflow can be used for biological sequences based on categorical factors (i.e., combinatorial variants of a T7 promoter, ribosome binding site (RBS), N- and C-terminal amino acids). Finally, we use METIS to improve a complex metabolic network, the so-called crotonyl-CoA/ethylmalonyl-CoA/ hydroxybutyryl-CoA (CETCH)[39] cycle, a new-to-nature synthetic

$CO_2$-fixation cycle, comprising 17 different enzymes plus 10 different cofactors and components, which was shown to be (thermodynamically) more efficient compared to natural photosynthesis. Yet, the network's full kinetic potential had not been exploited, as efficient strategies to explore its combinatorial space had been lacking so far. Using METIS allowed us to improve productivity of the CETCH cycle by ten-fold with (only) 1,000 experiments, resulting in the most efficient $CO_2$-fixing in vitro system described to date. Overall, these results demonstrate the ability of our workflow for the optimization of various complex biological networks with minimal experimental efforts, providing multiple opportunities for the study and engineering of different biological systems in the future.

## Results

### Assessing the performance of different algorithms for our workflow.
We first tested which machine learning algorithm would perform best with a limited number of experimental data typical for a standard research lab setup. To that end, we took advantage of an existing dataset from a recent optimization of an E. coli extract-based in vitro TXTL system[31]. In their study, Borkowski et al. optimized cell-free protein production in E. coli lysate by varying 12 different factors including salts, energy mix, amino acids, and tRNAs, and measuring production yield of Gfp (produced by a plasmid expressing Gfp) as output. Altogether, the dataset encompassed around 1000 data points. We fitted the dataset to obtain a standard as a gold regressor (a reference model fitted on pre-existing experimental data to evaluate new algorithms) and divided it further into test and training sets, with 20% and 80% of data, respectively. While the latter set was used to train the model, the test set was used to validate the gold regressor (Methods).

We used the gold regressor to assess the performance of four different machine learning algorithms over 10 rounds of active learning (Fig. 1a). The tested algorithms included deep neural networks (DNN), multilayer perceptrons (MLP), linear regressors, and XGBoost gradient boosting, which all show different capabilities for a given problem set and its data sample size. Over 10 rounds of active learning with 100 data points in each round, XGBoost and linear regressors showed better performance (Fig. 1b) compared to DNN and MLP, which generally need larger datasets for training[40]. XGBoost outperformed linear regressors when fewer data points per round were used (Supplementary Fig. 1).

XGBoost is an improved random forest-type algorithm, working through gradient boosted decision trees[41] by aggregating and compiling sets of models. This algorithm is a sparsity-aware, fast, scalable as well as versatile model for handling tabular data with complex non-linear interactions[41]. These features make XGBoost a promising algorithm for machine learning applications on different biological systems with limited datasets. For our workflow, we therefore selected XGBoost, which has also been used for different biological applications previously[18,42,43]. To determine the minimum dataset required for optimization, we compared active learning rounds with 5, 10, 25, and 100 data points. Notably, a sample size as low as 10 data points still allowed sufficient yield optimization (i.e., in the scale of the original study[31]) within 10 learning cycles (Fig. 1b).

### Testing the workflow with minimal experimental work.
Having validated the workflow with an existing data set, we next sought to test it in a real-world experimental setup, simulating a situation in which the number of combinations that can be tested is limited by available equipment, readout and experimental cost. We chose (again) to optimize relative Gfp production (yield) in an E. coli

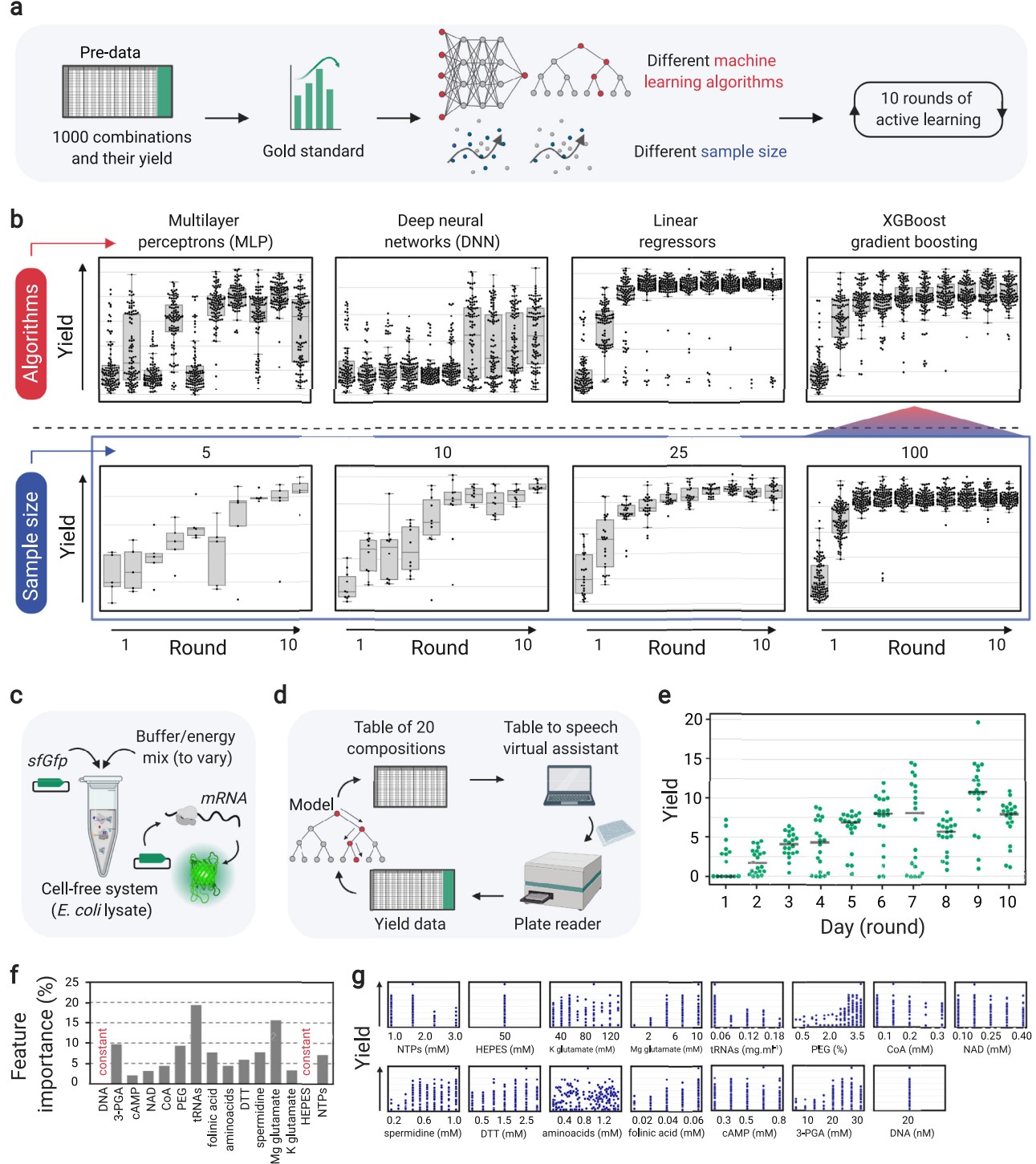

lysate TXTL system (Fig. 1c) that consists of 13 variable factors (components).

To optimize composition of the TXTL system, we defined a concentration range for each of the 13 factors (Code availability), and performed an active learning process over 10 rounds with only 20 experiments per round (Fig. 1d, see Supplementary Note 1 for details) quantifying Gfp yield (i.e., Gfp fluorescence reported from each composition normalized by the Gfp fluorescence of the standard composition[33]), as objective function. Over 10 rounds of active learning, the relative yield increased up to 20 and the median increased from zero to over 10 in the 9th round (Fig. 1e, see also Supplementary Fig. 2). Note

that low-yield data points (even those observed in the late learning cycles) are equally informative as high-yield ones, because they allow to explore the landscape around and beyond local maxima, as defined by the exploration to exploitation ratio of our workflow that we fully discuss in Supplementary Note 2.

Beyond the simple optimization of a given system, our workflow can also quantify the contribution of different factors during optimization. Figure 1f represents feature importance, i.e., the effect of each individual factor on the objective function. The importance is given as a relative fraction (or percentage) in the prediction of the values of the objective function by the model, with the sum of all factors set to 100%. Our analysis showed that

**Fig. 1 Assessing the performance of different algorithms and testing the active learning workflow with minimal data points. a** An existing dataset of cell-free gene expression compositions composed of 1000 data points was used to build a gold standard regressor and assess the performance of different machine learning algorithms in 10 rounds of active learning. **b** Top panel: performance of 4 algorithms, multilayer perceptrons (MLP), deep neural networks (DNN), linear regressors, and XGBoost gradient boosting in 10 rounds of active learning (100 data points per round). Bottom panel: performance of the XGBoost gradient boosting algorithm as the selected algorithm with different sample sizes. The boxplots with whisker length of 1.5, represent the minimum, 25th percentile (bottom bound of box), median (center of box), 75th percentile (upper bound of box), and maximum. **c** An in vitro or cell-free transcription-translation (TXTL) system (based on *E. coli* lysate) to test the workflow with 20 data points per round. A plasmid expressing *sfGfp* was added to TXTL reaction mix along with 13 components of reaction buffer and energy mix. **d** Overview of the active learning cycle. 13 components are varied starting with random compositions and over 10 rounds of results are imported to the model, which learns and suggests new compositions for improvement of the objective function. **e** The plot presenting the average of triplicates ($n = 3$ independent experiments) of the objective function (yield) for compositions in 10 rounds (days) of active learning. The gray lines show the median. **f** Feature importance percentages show the effect of each factor on the model's decision to calculate yields for the suggested compositions. **g** Distribution of different concentrations of each factor within the measured yields. The Google Colab Python notebook and all active learning data (combinations and yields) in this figure are available at https://github.com/amirpandi/METIS.

tRNA mix and Mg-glutamate were the most important components in optimizing Gfp yield, while cAMP and NAD were the least important contributors. Figure 1g shows the distribution of Gfp yield at different concentrations of individual factors (see also Supplementary Fig. 3). Decreasing concentrations of tRNA and NTP mixes correlated with high yield, while PEG 8000, Mg-glutamate, 3-PGA, folinic acid, and spermidine showed similar effects at increasing concentrations. Together, these data did not only result in an optimized TXTL system but also allowed to identify the most crucial components during system optimization, providing the basis for a deeper understanding of the system itself. All combinations and yields are provided as results files for each experimental round (**Data availability**), and the Google Colab notebook with all analyses and visualization modules are also accessible (Code availability).

**Development of METIS, a user-friendly, versatile modular workflow.** After demonstrating that our workflow is capable of working efficiently with minimal datasets, we sought to build METIS, a modular architecture that can be easily applied for the optimization of different biological objective functions. We implemented our workflow in Google Colab Python notebooks that can be accessed by the user—without installation or registration—simply through a personal copy of the notebook from a web browser. Defining the objective function and the variable factors (Fig. 2a), the user can simply open the link of Google Colab notebook and directly use the workflow as shown in Fig. 2a, b, Supplementary Figs. 4–6.

In Supplementary Note 2, we provide a detailed description of all features of METIS. The modular workflow enables the use of factors with numerical values (examples in Figs. 1, 3 and 5), categories (examples in Fig. 4, Supplementary Fig. 19), or both (example in Figs. 3 and 5). Active learning can be initialized by random combinations generated by the workflow in the first round (example in Figs. 1, 3 and 4). Alternatively, pre-existing datasets can be imported and used for optimization or simulations (examples in Supplementary Figs. 18 and 19). Although our workflow is designed as an active learning approach over iterative experimental rounds, it can be also used in a classical machine learning setup, when only using one round of experiments. Multiple data analysis and visualization modules are available that can be used in each round of active learning as shown in example applications (Fig. 2b, Supplementary Note 2). The workflow is able to generate a pipetting table output (exemplified for the experiments in Figs. 1 and 3), which alongside our table-to-speech virtual assistant tool, improves the speed and accuracy of manual pipetting (Supplementary Note 1, 2). For more complex experiments where multiple components in different volumes are required, the workflow can be interfaced

with lab automation (e.g., an Echo® acoustic liquid handling robot, see optimization of the CETCH cycle in Fig. 5).

**Application of METIS for optimization of a *LacI* gene circuit.** Next, we aimed to apply METIS for optimization of *LacI*-based gene circuits that were described recently[38]. Greco et al. developed a strategy for stringent gene expression by engineering transcriptional and/or translational small RNA inhibitors upstream of a *Gfp* reporter gene under the control of the pTAC promoter (Fig. 3a). Starting from a standard pTAC architecture, a so-called single-level controller (SLC), Greco et al. constructed three different multi-level controllers (MLC): pTHS (toehold switch; translational control), pSTAR (small transcription activating RNA; transcriptional control), and pDC (double controller; transcriptional and translational control)[38]. Notably, the authors could improve the rate of in vitro protein production by 35-fold with different MLC designs. Yet in these efforts, the fold-change in total protein production remained low (Supplementary Fig. 7), which was likely the result of leaky repressor-regulated promoters in the OFF state, as noted earlier[34,37]. A high fold-change in protein production, however, would be strongly desired for application of gene circuits, e.g., as diagnostic sensors, where a high signal-to-noise ratio is important. Additional to the high fold-change (FC), a desired circuit should have a high level of protein production, a feature that can be quantified by the dynamic range (DR) (Fig. 3a).

Here, we aimed at using our workflow to optimize the SLC and MLC *LacI* circuits. We performed 10 rounds of active learning with the objective function of FC × DR, to score those compositions that result in not only high fold-changes but also total Gfp productions. The fold-change can be improved by supplying an additional plasmid expressing *LacI* (under the control of a T7 promoter transcribed by purified T7 RNA polymerase) and the dynamic range can be improved through alternative selection of SLC and MLC circuits and through tuning TXTL composition. The active learning cycle received input from several factors in the *E. coli* cell-free system; amino acids and tRNAs, which are important when extra DNA is added, DTT as reducing reagent, spermidine for DNA-protein binding, and PEG 8000 as crowding agent (Fig. 3b), and four *LacI* circuits (one SLC and three MLC) were considered as one categorical feature with four alternatives. While the objective function improved during the active learning cycle (bottom plot in Fig. 3c), we did not observe a substantial improvement in fold-change of Gfp production alone (upper plot in Fig. 3c). Feature importance analysis identified the concentration of the $P_{T7}$-*LacI* plasmid as strong contributor (Fig. 3d, e, Supplementary Fig. 8), indicating deleterious LacI protein-DNA interactions or resource limitation of the TXTL system through production of the lacI protein[44].

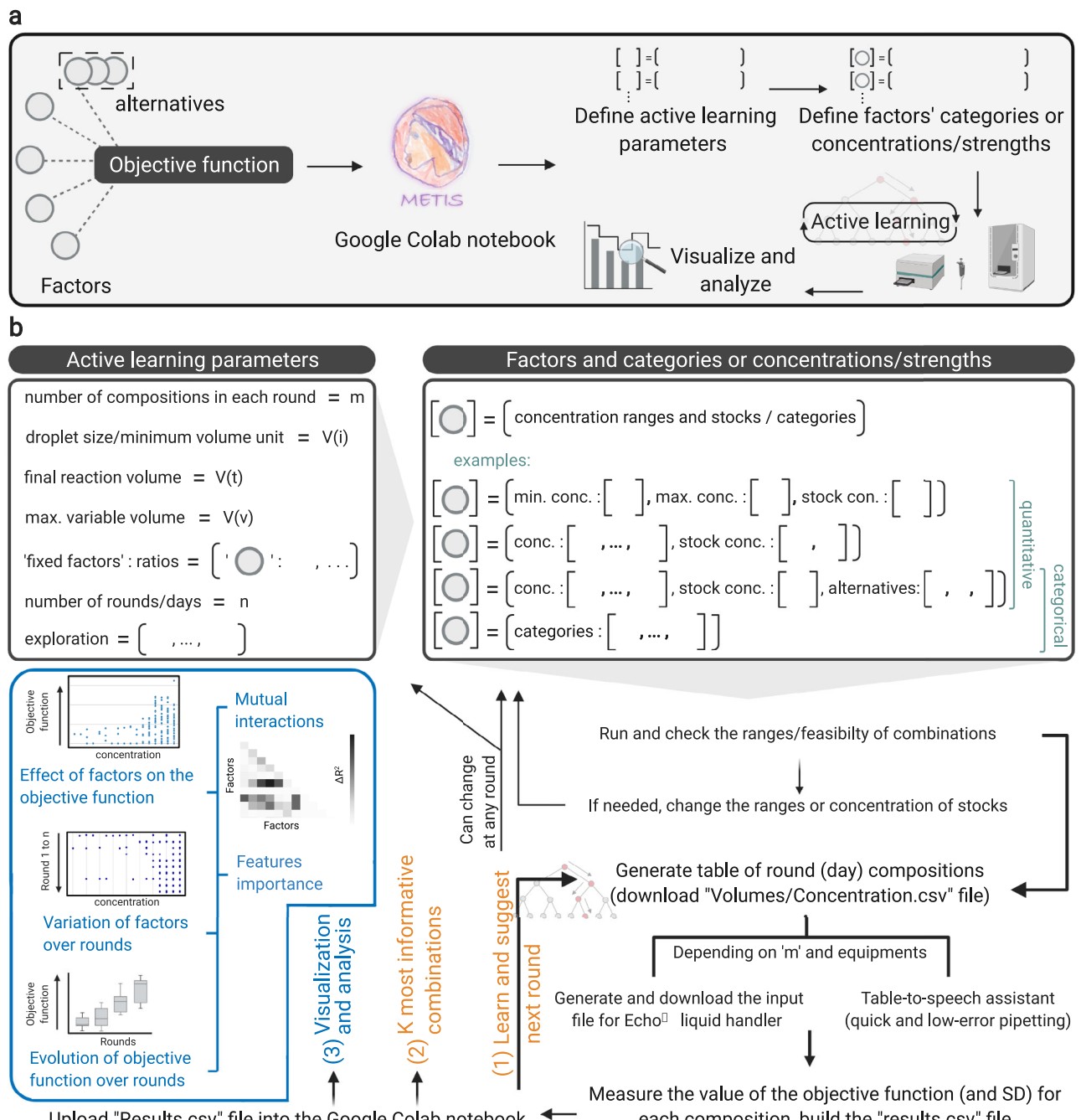

**Fig. 2 A representation of METIS, a modular active machine learning workflow for biological systems. a** The first step is choosing an objective function (an output/target that depends on multiple factors), then continuing with the Google Colab Python notebook, performing experiments, and visualizing and analyzing results. **b** Users should define active learning parameters depending on the application, equipment, and the size of the combinatorial space. Factors' ranges/categories are conditions that are varied to explore the behavior of the objective function. In each round of active learning, while the users perform experiments and label the suggested combinations with measured objective function values (parameters and factors' conditions can be readjusted at any round), the data can be analyzed and visualized using the workflow's modules. See Supplementary Note 2 and Supplementary Figs. 4–6 for a detailed explanation and guide for each step.

By performing a titration experiment with $P_{T7}$-*LacI*, we could show that addition of the *LacI* plasmid has indeed a strong negative effect on the optimal *LacI* circuit (i.e., with pTHS) (Fig. 3f) and in an independent TXTL protein production (Fig. 3g) (see also Supplementary Note 3 for details of the active learning cycle and titration experiments). To further investigate this effect, we titrated the *LacI* plasmid with either T7 or a constitutive promoter against a fixed concentration of the *Gfp*

expressing plasmid under control of either T7 or a constitutive promoter. While increasing concentrations of the plasmid with constitutive *LacI* expression did only slightly affect *Gfp* expression from the T7 promoter, increasing concentrations of *LacI* plasmid under T7 control strongly affected Gfp production, especially when Gfp was expressed from the constitutive promoter (Fig. 3h). These results indicated a resource competition between the two plasmids, according to which the T7 promoter wins competition

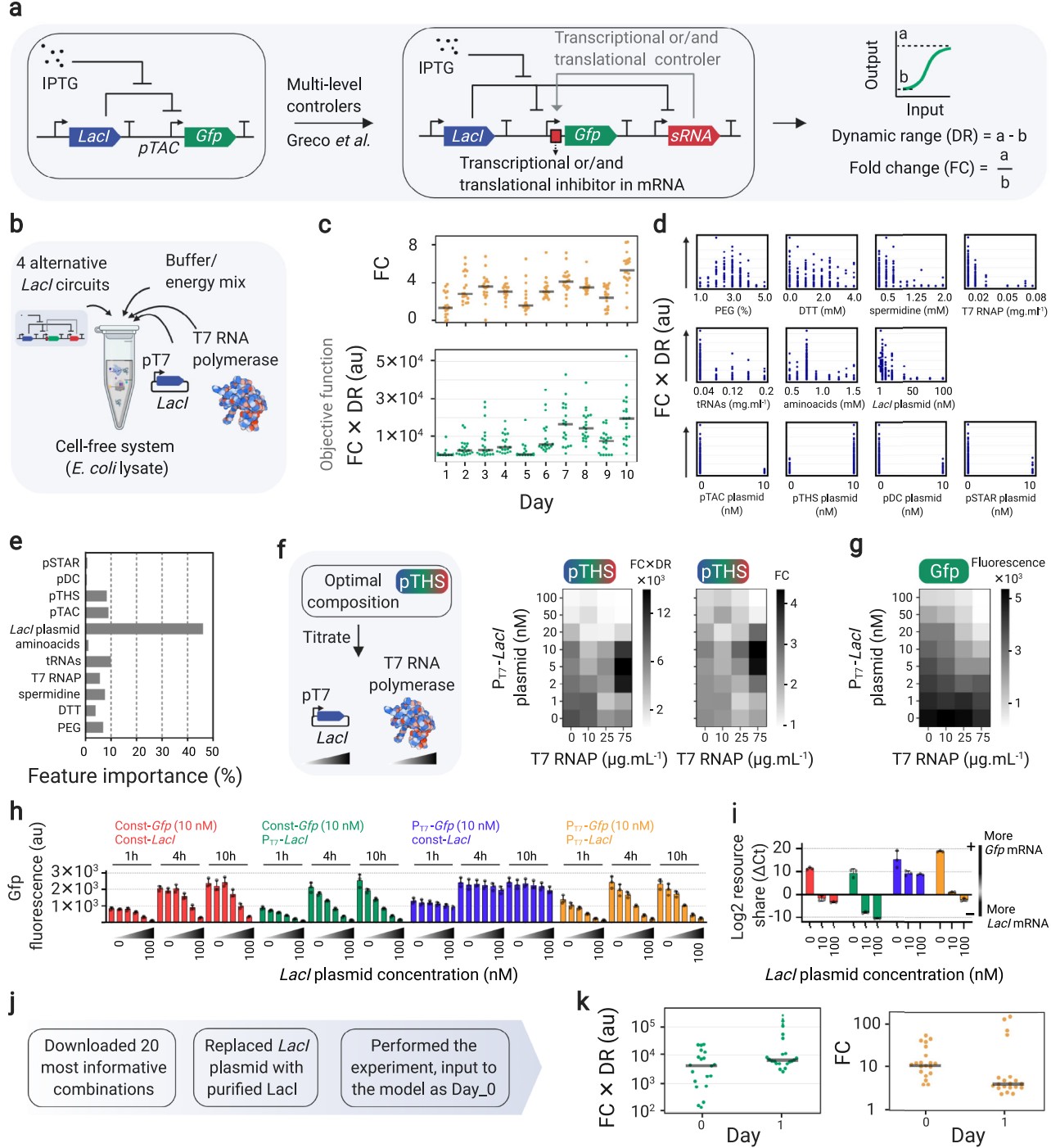

at the transcriptional and consequently the translational level. Quantifying the levels of *Gfp* and *LacI* mRNA by qPCR confirmed a direct correlation between mRNA and Gfp production levels, further supporting the resource competition hypothesis (Fig. 3i).

To overcome resource competition, we tested purified LacI protein instead of the *LacI* plasmid in the TXTL system, which resulted in improved Gfp productivity (Supplementary Fig. 9). Thus, we sought to optimize Gfp fold-change with using purified LacI protein instead of a *LacI* expressing plasmid. Using a module of METIS called "K most informative combinations" (with the number K to be defined by the user), we extracted the 20 most informative combinations of the active learning cycle and

repeated these 20 setup by replacing P$_{T7}$-*LacI* plasmid with purified LacI protein (Fig. 3j), resulting in a strong improvement in the objective function, and in particular Gfp fold-change (Fig. 3k). Note that among these 20 combinations were again all four SLC and MLC circuits. All of them improved upon providing external LacI, clearly demonstrating that resource competition had been limiting performance of the SLC and MLC circuits. Continuing with only one additional round of active learning using this dataset, we were able to improve the fold-change to up to 123 (Fig. 3k), which is 15-fold improvement compared to that of 10 rounds of active learning with the P$_{T7}$-*LacI* plasmid and 34-fold improvement compared to the initial setup.

**Fig. 3 Application of METIS for optimization of a *LacI* gene circuit. a** *LacI* gene circuits characterized by dynamic range (DR) and fold-change (FC) of the output (Gfp fluorescence) between 0 and 10 mM IPTG. **b** Active learning by varying components of *E. coli* TXTL, 4 *lacI* circuit plasmids as alternatives, T7 RNA polymerase and a T7-*lacI* plasmid. **c** The objective function (FC × DR) and fold change (FC) values, average of triplicates ($n = 3$ independent experiments) in 10 rounds of active learning. The gray lines show the median. **d** The distribution yield values within the range of each factor. **e** Feature importance percentages showing the effect of each factor on the objective function. **f** Titration of P$_{T7}$-LacI plasmid and T7 RNA polymerase with the optimal composition (from active learning that achieved with pTHS circuit). The heatmaps show FC × DR (left) and FC (right) values (average of triplicates, $n = 3$ independent experiments) of the titration. **g** Fluorescence values (average of triplicates, $n = 3$ independent experiments) of the similar titration as in **f** but instead of the pTHS circuit, a *Gfp* expressing plasmid was used). **h** Titration of *LacI* plasmids with constitutive/T7 promoter in combination with a *Gfp* plasmid with constitutive/T7 promoter. **i** The RT-qPCR results of the relative level of *LacI* and *Gfp* mRNAs after 10 h. Relative log2 resource share between *LacI and Gfp* mRNA in each sample is reported to account for RNA purification efficiency variability. In **h** and **i** bars are the average of triplicates ($n = 3$ independent experiments) and error bars are standard deviation. **j** Usage of the METIS module, K most informative combinations for further *LacI* circuit optimization. **k** Objective function FC × DR and FC (average of triplicates, $n = 3$ independent experiments) of 20 most informative combinations with purified LacI (Day 0) followed by Day 1 experiments suggested by METIS. The gray lines show the median. The Google Colab Python notebook and all active learning data (combinations and yields) in this figure are available at https://github.com/amirpandi/METIS. Source data for **f–i** are provided as a Source Data file.

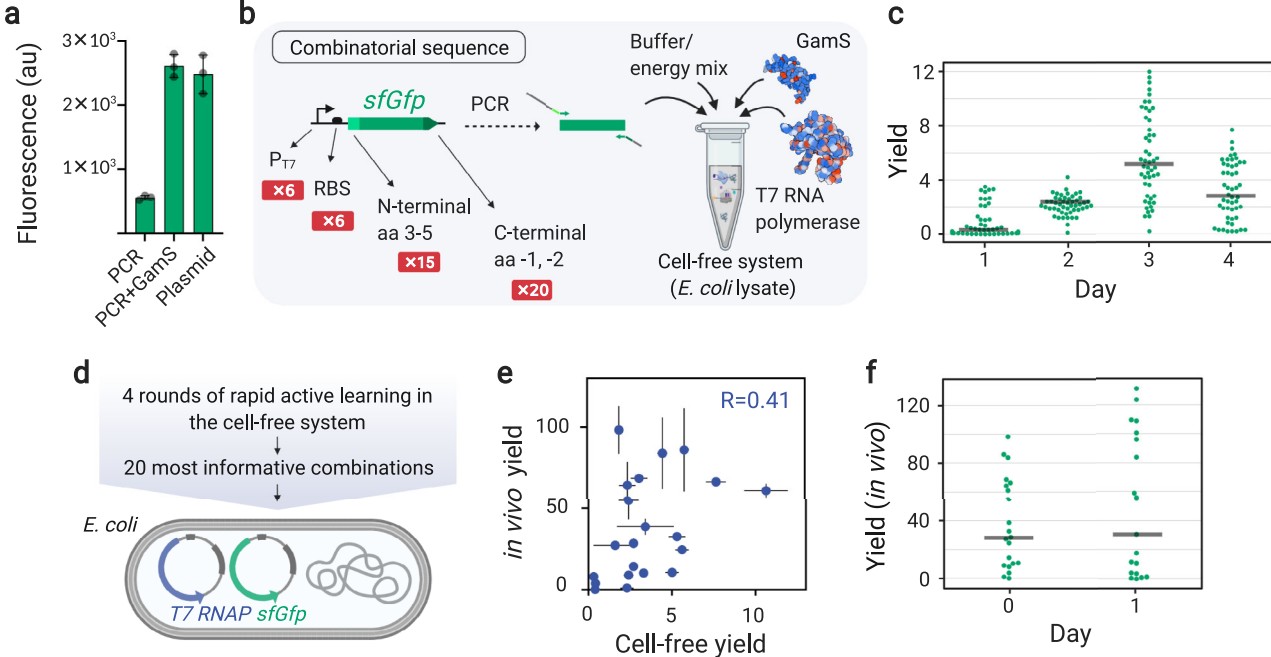

**Fig. 4 Application of METIS for optimization of a transcription & translation unit. a** The cell-free expression of *sfGfp* (super-folder *Gfp*) using plasmid, linear DNA (PCR) and linear DNA plus GamS protein, a nuclease inhibitor that protects linear DNA from degradation. The bars and the error bars are the average and standard deviation of triplicates ($n = 3$ independent experiments), respectively. **b** Design of a transcription & translation unit controlled by variants of a T7 promoter, ribosome binding site (RBS), N-terminal amino acids 3, 4, and 5, and the last two C-terminal amino acids. The combinatorial transcription & translation units are expressed from linear DNA in the TXTL system consisting of the *E. coli* lysate, buffer and energy mix, as well as purified GamS and T7 RNA polymerase. **c** The plot representing the average of triplicates ($n = 3$ independent experiments) as the result of 4 rounds of active learning, with 50 transcription & translation units tested per round. The yield is the Gfp fluorescence readout after 6 hours at 30 °C normalized by the same value from the reference constructs commonly used in the lab (Methods). The gray lines show the median. **d** A list of 20 most informative combinations of 4-day active learning performed in the cell-free system (c) was downloaded and the combinations were cloned in a vector and transformed into *E. coli* DH10β harboring a plasmid expressing auto-regulated T7 RNA polymerase (Methods). **e** Cell-free versus in vivo yields (average and standard deviation of triplicates, $n = 3$ independent experiments) for the 20 most informative combinations. **f** In vivo yield results (average of triplicates, $n = 3$ independent experiments) of Day 0 (20 most informative combinations) and Day 1 (suggested by the workflow). The gray lines show the median. The Google Colab Python notebook and all active learning data (combinations and yields) in this figure are available at https://github.com/amirpandi/METIS. Source data for **a, e** are provided as a Source Data file.

Overall, these experiments demonstrated how our workflow can be used to improve the signal-to-noise-ratio of an existing in vitro gene circuit by two orders of magnitude. Notably, the feature importance module of METIS, which identified apparent bottlenecks (i.e., resource competition by the *LacI* plasmid) and the K most informative combinations module of the workflow were crucial for success. A Google Colab notebook and all combinations and results are provided through Code and Data availability.

**Application of METIS on a transcription & translation unit.** To demonstrate that our workflow can also be used with categorical factors such as biological sequences, we tested METIS for the optimization of a transcription & translation unit. This unit is composed of six variants of a T7 promoter[45], six ribosome binding sites (RBS)[46], as well as 15 variations of N-terminal amino acids 3 to 5[47], and 20 variations of the last two C-terminal amino acids[48], which is in line with two recent studies that

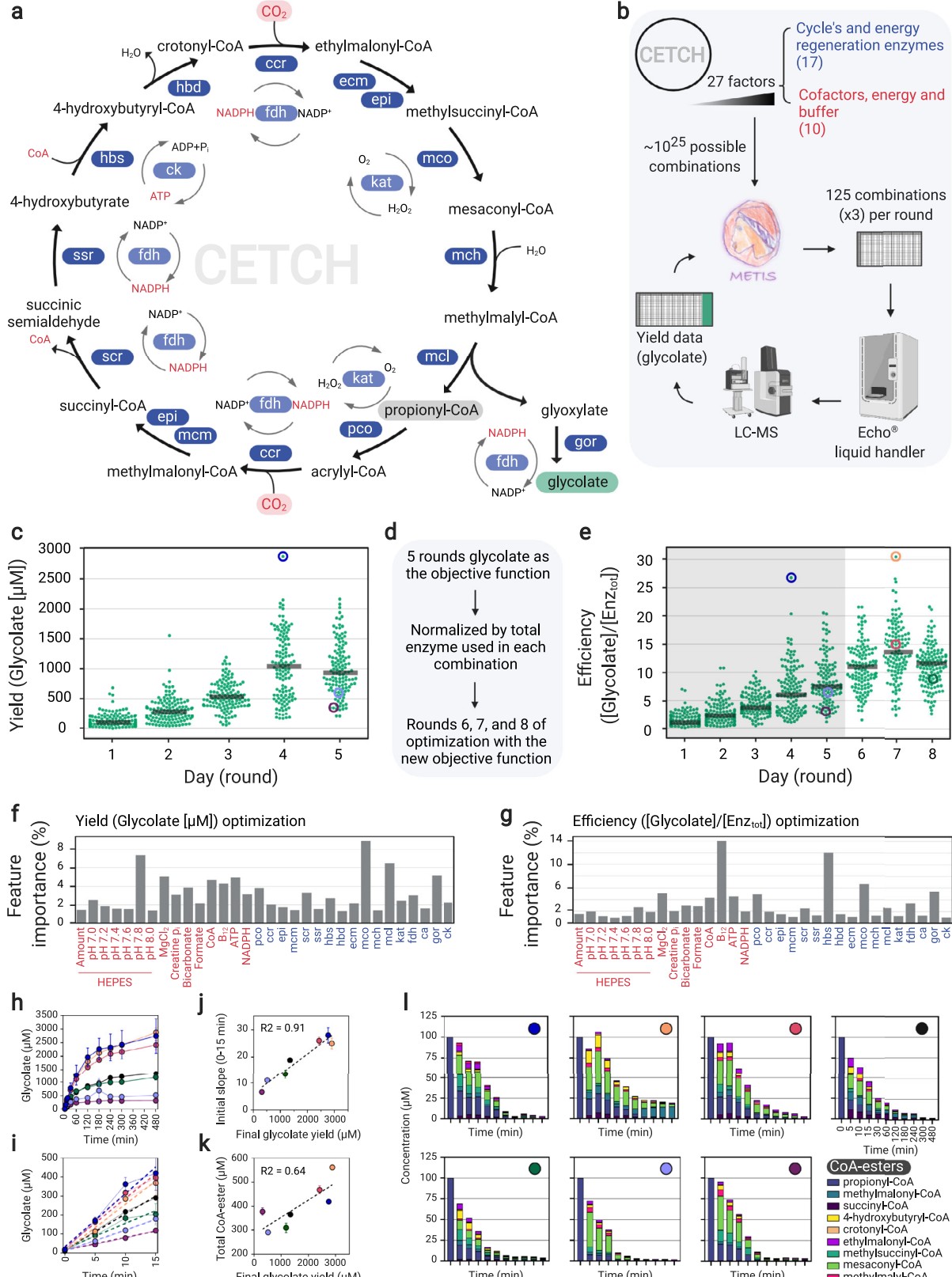

reported the importance of N- and C-terminal amino acids on mRNA translation[47,48].

To establish a convenient cell-free screening system, we sought to use linear DNA (i.e., a PCR product)[49] as template in combination with GamS, a small, 136 amino acid-long nuclease inhibitor from phage λ[50] that binds and protects linear DNA

from degradation. First, we validated that addition of linear DNA with GamS resulted in gene expression levels comparable to that of plasmid DNA (Fig. 4a), which allowed the fast and efficient assembly of DNA templates through PCR primers without extensive cloning, transformation, and plasmid preparation steps (Fig. 4b).

**Fig. 5 Application of METIS for optimization of an in vitro CO₂-fixation pathway (CETCH cycle). a** Reaction sequence of the CETCH cycle (see Methods for enzyme names and information). **b** Active learning with 125 conditions tested in each round. ECHO® liquid handler pipetted the combinations and the reactions were started with 100 µM propionyl-CoA and stopped after 3 h. The glycolate content was measured by LC-MS. **c** Optimization of the CETCH cycle with glycolate yield. **d** Summary of the optimization and the switch of the objective function. **e** Transformed data of **c** (glycolate yield divided by the total amount of enzymes = efficiency) for rounds 1–5, shaded region, and the data of three additional rounds of optimization with efficiency as the objective function (rounds 6–8). The yields in **c** and **e** are average of triplicates, ($n = 3$ independent experiments) and the gray lines show the median. **f, g** Feature importance of factors for active learning in **c** and **e**, respectively. **h–l** Manually pipetted experiments for seven conditions, three highest glycolate yields (blue, orange and red), a control (black) and three randomly picked underperformed conditions (green, lavender, burgundy) color coded the same in **h–l** and circled in **c** and/or **e**. These plots show glycolate production over 8 h (**h**) and its first 15 min with slopes (**i**), initial production rate versus the final glycolate yield (**j**), total amount of measured CoA esters after 8 h versus the final glycolate yield (**k**), and quantified CoA esters over 8 h (**l**). The plotted values in **h–k** are the average of triplicates ($n = 3$ independent experiments), and the error bars represent the standard deviation. In (**l**), bars are the average of triplicates ($n = 3$ independent experiments), each compound is plotted with error bars in Supplementary Fig. 17. In **l** the amount of propionyl-CoA within the zero samples is the added amount (100 µM) to start the reaction and was not measured by LC-MS. The Google Colab Python notebook and all active learning data (combinations and yields) in this figure are available at https://github.com/amirpandi/METIS. Source data for **h–l** are provided as a Source Data file.

We then optimized the transcription & translation unit that theoretically consists of 6 ($P_{T7}$) × 6 (RBS) × 15 (N-terminal) × 20 (C-terminal) = 10,800 potential conditions (i.e., combinations) through screening of only 200 combinations in 4 rounds of active learning (Fig. 4b). As the objective function, we defined the yield of the Gfp fluorescence readout of each transcription & translation unit normalized by a construct comprising wild-type T7 promoter, B0032 RBS and *sfGfp*. Yields were quantified after 6 hours of incubation of the different transcription & translation units at 30 °C in the *E. coli* cell-free system supplemented with purified GamS and T7 polymerase. Over 4 rounds of active learning, yield of the transcription & translation unit improved up to 12-fold on Day 3. Using a high exploration rate on Day 3 resulted in a wide distribution of yields, but no further improvement, indicating that an optimum had been reached (Fig. 4c). The distribution of alternative factors within the yield of 200 combinations and a representation of the feature importance are shown in Supplementary Fig. 10. Altogether, our experiments demonstrated again, how METIS can be used to improve a described genetic unit by more than an order of magnitude with minimal experimental efforts.

After having rapidly explored the combinatorial space of the sequence controlling the transcription & translation unit in a cell-free setup, we additionally investigated the effect of the 20 most informative combinations in vivo (Fig. 4d). Surprisingly, however, the cell-free and in vivo yields for the 20 combinations showed a relatively low correlation of 0.41 (Day 0, Fig. 4e, Supplementary Fig. 11). This indicated that although cell-free systems offer rapid prototyping solutions, the optimal candidates are not necessarily directly transferable in vivo. To investigate whether we can further improve the performance of the transcription & translation unit in vivo, we used the data from Day 0 and continued with one more round of experiments guided by our workflow (Day 1, Fig. 4f). This resulted in an improvement by 130% for the highest yield in vivo.

**Application of METIS for optimization of the CETCH cycle.** Finally, we aimed at assessing the performance of METIS for the optimization of complex metabolic networks. The collection of thousands of different enzymes and recent progress in enzyme engineering has opened the way for the design and construction of synthetic metabolic networks with new-to-nature properties[35,51,52]. One recent example is the CETCH cycle (Fig. 5a), a synthetic in vitro metabolic network consisting of 17 different enzymes that was built around a highly efficient CO₂-fixing enzyme, Crotonyl-CoA carboxylase/reductase (Ccr), converting CO₂ into the C2-compound glyoxylate[39] or glycolate[53]. Notably, the CETCH cycle is more efficient than natural

occurring CO₂-fixing pathways like the Calvin-Benson-Bassham (CBB) cycle[39]. However, since the enzymes used for its construction derive from different organisms and thus metabolic backgrounds, several rounds of rational optimization were needed to harmonize the enzyme reactions and cofactors used in the cycle; and even though the kinetic parameters of the individual enzymes are known, their interactions in such a complex setup are non-linear, hardly predictable and basically impossible to disentangle with pure rational approaches. Hence, we sought to use our active learning workflow to improve the CETCH cycle's productivity further.

The setup of the CETCH cycle consists of 27 components encompassing 13 core enzymes, as well as four accessory enzymes, and nine other components such as magnesium chloride, CoA, NADPH, ATP and the starting substrate propionyl-CoA (see all components in Fig. 5 and their concentration range in the Code availability). To minimize handling errors and automate the experimental setup of individual CETCH assays, we used an ECHO® 525 acoustic liquid handler with a minimal pipetting volume of 25 nL. Miniaturizing the assay to 10 µl of total volume allowed us to work with 384-well plates and assay 125 different conditions in triplicates per active learning round (Fig. 5b). To determine the CETCH cycle's productivity (i.e., formation of glycolate from CO₂), we developed an LC-MS (liquid chromatography-mass spectrometry) method using ¹³C₂-glycolic acid as an internal standard (Methods).

For the first five rounds of optimization, we used product yield (glycolate) as objective function (for a description of the used parameters see Supplementary Note 4). After four iterative rounds, we reached a final concentration of 2.87 ± 0.09 mM glycolate in the best performing condition starting from 100 µM propionyl-CoA (Fig. 5c). This yield translates into 57.4 fixed CO₂-equivalents per acceptor (propionyl-CoA) and is >10 times more productive compared to the originally reported, already rationally optimized version 5.4 of the CETCH cycle[39].

As we had not restricted the component resources during optimization, most of the superior conditions used more enzymes (compared to CETCH 5.4) to increase glycolate production (Supplementary Fig. 12). Next, we aimed at increasing specific productivity of the CETCH cycle. To that end, we took the data from the initial five rounds of unrestricted optimization and divided the glycolate yield values by the total concentration of enzymes used for each combination. This data was fed back to METIS and three additional rounds of active learning were performed with the new objective function, called "efficiency" (Fig. 5d). Optimization of efficiency identified one condition in round seven that is about six times more efficient than CETCH 5.4 and 14% more efficient than the best condition from the

unrestricted optimization achieved in round four (Fig. 5e, see also Supplementary Figs. 12 and 13).

To learn more about the possible bottlenecks of the CETCH cycle, we used the feature importance module of the METIS workflow along with plots visualizing the yield distribution over the range of each factor (Supplementary Figs. 14, 15). One of the most important contributors for both optimization efforts is the enzyme Methylsuccinyl-CoA dehydrogenase (Mco) (Fig. 5f, g). The enzyme's low activity of 0.1 U/mg and its unstable substrate methylsuccinyl-CoA, which is prone to spontaneous hydrolysis, likely require large amounts of Mco to preserve flux through the cycle[54]. During efficiency optimization, the two most important components were 4-hydroxybutyryl-CoA synthetase (Hbs) and coenzyme $B_{12}$ (Fig. 5g). Analysis of the top 10% best performing conditions (Supplementary Figs. 12 and 13) revealed that the concentrations of Hbs and $B_{12}$ were significantly lower compared to the control (CETCH 5.4). To verify that high concentrations of Hbs have a negative impact on the cycle, we tested our control assay with ten times less and with five times more of the enzyme. Indeed, increasing Hbs concentration in the original assay decreased yield by 40%, while decreasing Hbs by one order of magnitude did not lower glycolate yield (Supplementary Fig. 16). Regarding the negative impact of higher concentrations of $B_{12}$, we reasoned that cobalt released from damaged cofactor could inhibit enzymes. Similar to high concentrations of Hbs, addition of cobalt to the original assay led to a decrease in glycolate yield (Supplementary Fig. 16).

To understand the dynamic behavior of the different CETCH cycle variants, we manually repeated the top three conditions (highest glycolate yields), a control (see Supplementary Note 4) and three underperforming conditions, taking time point samples for eight hours. The yield from this manual approach reflected the yield from the previous automated, miniaturized experiments, validating the results of our optimization efforts (Supplementary Table 3). Interestingly, the final glycolate yield after eight hours (Fig. 5h) and the initial glycolate formation rates of these conditions over the first 15 minutes (Fig. 5i) were highly correlated (Fig. 5j), indicating that total flux and not improved enzyme/cofactor stability (or life-time) was responsible for the observed increased productivity of the system. This trend was further confirmed by a detailed analysis of 9 CoA-ester intermediates at different time points (Fig. 5k, l). Quantification of the CoA-ester intermediates did not show accumulation of single metabolites in the underperforming conditions or the control, indicative of specific bottlenecks (Fig. 5l, Supplementary Fig. 16). Instead, the underperforming conditions showed overall a faster depletion of intermediates, in line with the hypothesis that high flux through the cycle is important to prevent the loss of intermediates towards side reactions or hydrolysis.

In summary, our optimization efforts of the CETCH cycle resulted in variants that showed more than ten-fold productivity and almost six-fold improved efficiency, representing to the best of our knowledge the most efficient in vitro CO2-fixing system described to date.

## Discussion

In this work, we describe METIS, a versatile, modular active learning workflow for the optimization of various biological objective functions, such as genetic and metabolic networks. This study democratizes machine learning applications for experimentalists without any programming skills or sophisticated lab equipment. We provide Google Colab notebooks (see Code availability) that can be adapted to different optimization applications (also known as Bayesian optimization) and even used for data-driven predictions (for use of the latter see Supplementary Table 1, Supplementary Note 5, Supplementary Fig. 18).

For tailoring the workflow, the number of rounds and experiments per round need to be defined, which should take into account the number of different factors and their conditions, complexity of the objective function, as well as experimental throughput. For applications with a larger combinatorial space, more combinations need to be tested (Fig. 5). However, if the number of experiments is limited by cost, effort, or lab equipment, performing active learning in more rounds can be used to compensate for a lower number of total combinations tested. To explore a system beyond a local optimum, it is advised to adapt the exploration to exploitation ratio for each round individually (fully discussed in Supplementary Note 2). Users should apply their knowledge on the system and implicitly check whether the value of a given factor is fixed too early, probably indicating a low exploration to exploitation ratio. On the other hand, a high exploration to exploitation ratio might push the model towards random combinations, asking for a proper balance to enable explorative as well as exploitation sampling. In our empirical experience, the exploration to exploitation ratio should gradually decrease towards the late rounds of active learning to enable more explorative combinations in early rounds and more exploitation in late rounds for efficient optimization (Supplementary Note 2).

Workflows can be started either from scratch (random combination as initialization) or using existing datasets (then performing active learning). Although our workflow is designed as an active learning approach (over multiple rounds of experiments), it can also be used as a classical machine learning with only one round of experiments. Factors of a given objective function can be numerical and/or categorical. Active learning parameters can be further customized using a detailed explanation in Supplementary Note 2.

METIS provides a variety of choices for visualization and analysis of results. Most importantly, our workflow can quantify importance of individual features and provide a number of most informative combinations, which has both proven particularly useful during LacI gene circuit optimization (Fig. 3). Using these features of the workflow allowed us to not only to improve the fold-change of the circuit, but also spot and, using additional experiments, verify a major bottleneck in the further optimization of the system (i.e., the LacI expression plasmid). After replacing the LacI expression plasmid with purified LacI protein, we were able to improve the circuit by more than two orders of magnitude compared to the original system. Notably, we did not have to re-perform active learning when switching to purified LacI instead of the LacI plasmid. The 20 most informative combinations generated through our workflow offered a short and quick path toward optimization.

Applying METIS onto different biological systems, we demonstrate that our workflow is able to optimize several complex genetic and metabolic networks of medium to large combinatorial space with minimal experimental efforts. As example, we improved the CETCH cycle a system of 27 variable factors including enzymes, cofactors, and buffer composition, spanning a theoretical combinatorial space of $\sim 10^{25}$ different conditions. Performing only 1,000 (triplicate) assays over 8 rounds of active learning yielded a system with ten-fold improved productivity and six-fold increased efficiency, representing the most efficient in vitro $CO_2$-fixation system described to date.

The development and application of complex genetic and metabolic networks in synthetic biology is dramatically increasing and require new tools for their data-driven analysis. Efficient explorative approaches are needed not only for the optimization of existing biological networks, but also for the design and realization of new-to-nature genetic and metabolic networks for which sampling the entire combinatorial space becomes practically impossible. Apart from network optimization with minimal experimental datasets, METIS can simultaneously help to discover so far unknown interactions and bottlenecks in these

networks, which paves the way for their hypothesis-driven improvement. In the *LacI* circuit optimization, we showed how a bottleneck (i.e., resource competition) can be identified, targeted, and finally overcome, which allowed us to improve the system by 34-fold. Similarly, during optimization of the CETCH cycle, we identified Mco, Hbs and $B_{12}$ as limiting factors.

Numerous applications of the METIS workflow can be envisioned in the future, including the optimization of growth media and/or biochemical assays, genetic circuits, from simple transcription & translation units to more complex designs, or the guided engineering of proteins, enzymes, and metabolic pathways in vivo and in vitro. With its convenience and easy access, METIS opens the door for the study, prototyping, (combinatorial) engineering, and optimization of these systems in an efficient, standardized, and systematic manner.

## Methods

**Gold regressor and analyzing different machine learning algorithms**. To find out which machine learning algorithm and sample size are suitable for our workflow, we conducted the following simulation:

- 1017 data points (compositions and yields) were collected from a recent study[31].
- An XGBRegressor model (gold regressor) was trained on 80% of the dataset and 20% of the dataset was used for validation and to avoid overfitting via early stopping.
- 100 combinations produced randomly within the range of each factor for Day_1.
- Instead of doing experiments in the laboratory to determine the yield of each combination, yield values were assigned by the gold regressor.
- Note that, in this phase the test model predicts the yields and ranks them to suggest for the experiments of the next day, and the gold regressor (trained on pre-data) is used to assign yield values by prediction instead of performing the experiments in the laboratory.
- For each machine learning model (MLP, DNN, linear regressors, XGBoost) an ensemble of 5 models with different hyperparameters was produced.
- Note that the linear regressors is a deterministic approach so we just duplicated a model 5 times for which all predictions are the same.
- Each ensemble was trained on Day_1 data.
- 100000 random combinations were generated, and their corresponding yield was predicted by the ensemble of models and ranked by UCB score (see method section for the core algorithm of active learning), top 100 combinations were suggested for the next day. Yields were assigned by the gold regressor.
- The last two steps were repeated for other days, and on each day the model was trained on all the previous days' data.

Note that, in Fig. 1b for different sample sizes with XGBoost, 5, 10, 25, or 100 combinations were suggested for the next day.

Hyperparameters: MLPRegressor from Sklearn (fully connected architecture with Relu activation function) was used for MLP. In ensemble of 5 models the following number of neurons were used in the hidden layer: (10, 100, 100, 20), (20, 100, 100, 10), (20, 100, 100, 20), (10, 100, 100, 10), (20, 100, 100, 50). For DNN we used the Keras implementation of fully connected layer architecture with 100, 100, 20 neurons for each of hidden layers. For Linear Regression the default implementation of Ordianry Least Square by Sklearn was used. XGBRegressor with following parameter was used for XGBoost model: objective = 'reg:squarederror', n_estimators = 500, learning_rate = 0.01, max_depth = 6, min_child_weight = 1, subsample = 0.8.

**General description of METIS notebook**. All scripts used in this study were written in Python 3. Our modular tool, METIS, runs on Google Colab working through web browsers with a link without users needing to install Python or any packages.

Packages used in the development of METIS:

- Data processing: pandas (1.1.4) and numpy (1.18.5)
- Data visualization: matplotlib (3.2.2) and seaborn (0.11.0)
- Machine learning and deep learning: scikit-learn (0.22.2.post1), xgboost (0.90), and Keras (2.3.1) using TensorFlow backend.

**The core algorithm of active learning**. After measuring the value of the objective function (yield) for random combinations of Day_1, we continued with the following algorithm:

- RandomSearchCV is used to find the optimal 20 hyperparameters for the XGBoost model.
- The ensemble of 20 models is trained with the hyperparameters on data from all previous days (Day_1 to present day).

- 100000 combinations out of possible combinations are randomly selected.
- The mean and standard deviation of ensemble predictions are calculated.
- The combinations are sorted based on Upper Confidence Bound (UCB) score:[31] exploitation * (average of predictions) + exploration * (standard deviation of predictions).
- To perform experiments of the next day, the combinations with the highest UCB values are suggested.

The high standard deviation represents the uncertainty and improves the prediction power of models, whereas a high average value weighs favorable combinations leading to higher yields. Hence a coupled score taking into account these two factors ranks the most promising combinations[31]. Note that the active learning for optimization of objective functions is also called Bayesian optimization[55]. In Supplementary Fig. 21 we show optional data preprocessing and an improved XGBoost model. See Supplementary Fig. 22 for an optional scoring (can be defined when using METIS), batch UCB that can generate richer combinations for subsequent rounds.

**Finding K most informative combinations**. The K most informative combinations are calculated using the following algorithm:

- RandomSearchCV is used to find the optimal 20 hyperparameters for the XGBoost model.
- 2000 subsets of length K are selected from the tested combinations. The total number of possible subsets is represented in Eq. (1).

$$\binom{N}{K} = \frac{N!}{K! \times (N-K)!} \tag{1}$$

- Then a new XGBoost with the optimal hyperparameter is trained on each subset. The model performance is then validated on unseen combinations using the Spearman correlation coefficient.
- All subsets are sorted based on their Spearman correlation coefficient, the top 5 are then chosen. Each of these 5 could be used.

Note that increasing the number of subsets leads to a longer training time.

**Finding feature importance**. Feature importance values have been calculated with the following algorithm:

- RandomSearchCV is used to find the optimal hyperparameter for the XGBoost model.
- The model is trained using the selected hyperparameter. Using the built-in "feature_importances_" property of the XGBoost package, the ratio of feature importance is calculated throughout the training process for each day cumulatively.

**Finding nonlinear (mutual) interactions**. In complex systems, factors usually interact with each other and epistatically affect the output. These interactions can be among many factors, however, the most relevant is the mutual or double interaction between factors[56]. This analysis can be a hint to discover biological phenomena's behavior. The mutual interactions were calculated through the following algorithm:[57]

- A linear regression model is fitted on the dataset and its performance is evaluated based on the R squared of predicted and actual values. This performance is considered as the baseline.
- Iteratively, a new feature is added to the temporary dataset that equals $F_i \times F_j$ for i and j in the list of factors.
- The linear regression is fitted on the temporary dataset (which now has one more feature, $F_i \times F_j$) its performance is measured similarly to the baseline.
- The difference between each performance and the baseline, j, is calculated and visualized.

**METIS prediction**. In contrast to METIS optimization that tries to find the most promising combinations through maximizing the objective function, METIS prediction aims to maximize the model performance on the prediction of the objective function for unseen combinations. We modified the core active learning algorithm:

- Instead of UCB (exploitation × mean + exploration × std), combinations are sorted based on only their std value and set exploitation to zero. This enables picking the most uncertain combination for the next round.
- At the end of each round, it returns a trained model instead of promising combinations, and the R squared of prediction is improved over rounds.

**Performance analysis using cross-validation**. To evaluate the model performance of the enzyme engineering notebook, we used k-fold cross-validation. In each round, all the tested combinations are divided into k subsets (k = 5 for Supplementary Figs. 18, 19), then in five steps we trained the model on 4 and evaluated its performance ($R^2$ Pearson) on the other subset. This process was

repeated for all 5 subsets. In the end, the average performance on all subsets was reported as the model's performance. We used sklearn built-in function for cross-validation.

**Table-to-speech virtual assistant**. This tool helps molecular biologists to boost their manual liquid handling through reading volume and destination well in ascending order, therefore minimizes the need for changing the pipetting volume. We used the Google Text2Speech python package to transform the text into a voice file. There are two ways to interact with this notebook to continue with the next pipetting volume. The first is to do it manually with your keyboard (what we did), the second is using the voice assistant. For transforming voice to text (specific commands like 'next', 'repeat', etc.). We used the SpeechRecognition (3.8.1) python package. The code is available on https://github.com/amirpandi/Liquid-Handling-Assistant.

**Plasmid and DNA preparation**. The constitutive *Gfp* under the control of J23101 promoter and B0032 RBS was built in a recent study (pBEAST-J23101-B0032-*sfGfp*)[58]. Using golden gate cloning (BsaI-HF®v2 NEB #R3733L, T4 DNA ligase NEB #M0202T), in this plasmid, the super folder *Gfp* gene was replaced by *LacI* for constitutive-*LacI*, then the promoter was replaced by a T7 promoter (gaattttaa-tacgactcactatagggaga) to construct $P_{T7}$-*LacI* plasmid. Since we used T7 promoters, a T7 terminator from Temme et al.[59] (tactcgaacccctagcccgctcttatcgggcggctaggggtttttttgt) was cloned downstream. The version of *LacI* gene is similar to those in *LacI* circuits built by Greco et al.[38]. Plasmids for the cell-free gene expression were purified using the Machery-Nagel NucleoBond Xtra Maxi kit. For protein purification using His tag, *sfGfp* and *LacI* genes were cloned with an N-terminal His tag under IPTG-inducible T7 promoter.

For cell-free experiments for optimization of the transcription & translation unit (Fig. 4b), PCRs were performed using Q5® High-Fidelity 2X Master Mix (NEB #M0492L), *sfGfp* as the template, and primers (Eurofins and Sigma-Aldrich) with overhangs harboring $P_{T7}$, RBS, and N-terminal sequence (forward primer) and C-terminal (reverse primer) at the final volume of 50 μL. After verification of PCRs using agarose gel, Monarch PCR & DNA Cleanup Kit (NEB #T1030L) was used to purify the fragments and they were all adjusted to the concentration of 100 nM to use for active learning experiments.

For in vivo experiments of the transcription & translation unit (Fig. 4d) PCRs were done similar to the cell-free experiment. Restriction sites for BsaI enzyme were designed on either side of PCR fragments enabling for goldengate assembly into a pSEVA224 vector (a low copy plasmid with kanamycin marker) from the SEVA collection[60,61]. Since we used T7 promoters, a T7 terminator from Temme et al.[59] (tactcgaacccctagcccgctcttatcgggcggctaggggtttttttgt) was cloned downstream.

**Protein purification**. For all enzymes involved in the CETCH cycle, expression and purification were performed as previously described[62]. Other proteins, T7 RNA polymerase (addgene #124138), GamS (addgene #45833), sfGfp, and LacI were His-tag purified using Protino® gravity columns (Machery-Nagel #745250) and Protino® Ni-NTA Agarose (Machery-Nagel #745400). 1 L cultures in LB media supplement with appropriate antibiotic were subcultured (1:100) from overnight precultures. Cultures were grown at 37 °C for two hours, then induced by 0.1 mM IPTG, incubated for 3 more hours at 37 °C to produce proteins. Cells were harvested at 8000 g for 10 min, pellets were resuspended with 5 mL NPI-10 buffer, and sonicated. Samples were centrifuged at 18000 g for 1 hour at 4 °C. The equilibration, wash, and elution steps were done according to the manufacturer's protocol. Next, imidazole desalting was performed using PD-10 desalting columns (GE Healthcare #17085101) according to the manufacturer's protocol. The purification was verified using the SDS page and the protein concentrations were determined using the Bradford assay. Glycerol was added to the protein samples to a final percentage of 10%, then they were aliquoted and after flash-freezing in liquid nitrogen, stored at −80 °C.

**Lysate preparation**. E. coli lysate was prepared using an autolysis strategy[63]. Briefly, freeze-thawing E. coli BL21-Gold (DE3) cells with a pAS-LyseR plasmid produce a high-quality extract. Overnight precultures in LB-ampicillin media at 37 °C were subcultured in 5 × 2 L 2xYTPG medium supplemented with ampicillin and grown at 37 °C to the OD = 1.5. Cells were harvested (2000g, 15 min, room temperature) in 10 centrifuge bottles and 90 mL of cold S30A buffer (50 mM Tris-HCl at pH 7.7, 60 mM K-glutamate, 14 mM Mg-glutamate, to the final pH of 7.7) was added to each. After vigorous vortexing, each was divided into two preweighed 50 mL falcons and centrifuged (2000g, 15 min, room temperature). The supernatants were removed carefully and after weighing falcons with pellets, the net weights were calculated. Two volumes of cold S30A with 2 mM DTT, were used to resuspend each pellet (2.8 mL for 1.4 g pellet), which were then vortex-mixed, and stored at −80 °C. The next day, frozen cells were thawed in a water bath at room temperature, vigorously vortex-mixed, and incubated at 37 °C shaking for 45 min. The vortexing and 45 min incubation steps were repeated. Finally, the samples were centrifuged (30000 g, 60 min, 4 °C) to obtain the cell extract. The supernatants were gently pipetted out in 1.5 tubes, recentrifuged (20000 g in a tabletop centrifuge, 5 min, 4 °C) to remove all the remaining cell debris aliquoted, and after freezing in liquid nitrogen stored at −80. For the composition of the cell-free reaction buffer

and energy mix, all chemicals were used as by Sun et al.[33] except for amino acids (L-amino acids set, Sigma #LAA21-1KT).

**Cell-free reactions**. To perform the active learning experiments in Figs. 1 and 3, Table2Seech_Volume.csv file of each round was downloaded from the notebook and uploaded to the table-to-speech virtual assistant notebook. Before starting the pipetting, we arranged all pipette tips with numbers written on tip boxes (two boxes side by side) from 1 to 20 (for 20 data points). PCR tubes in which the compositions were going to be mixed also were numbered on racks from 1 to 20. The numbering increases the accuracy of the manual pipetting. Next, the table-to-speech assistant was run on a laptop on the bench and the space key was set in the Google Colab settings to run the code. After pipetting each factor into the corresponding destination, while the right hand was replacing the tip, the left hand pressed the space key to hear the next pipetting step in a headphone as well as to see the action appearing on the screen. The table-to-speech assistant goes line by line for each factor and ranks the pipetting values from minimum to maximum, hence, minimizes changes in the pipette volume. For fixed elements such as HEPES and lysate, a master mix was made and after finishing pipetting all combinations, the master mix was added to each. All the steps were performed on ice. At the end, samples were gently mixed (not to generate bubbles) using a multichannel pipette and 10 μL of each was transferred into a 384-well plate (Greiner Bio-One #784076). Note that the volume of mixtures should be at least 20% in excess in PCR tubes not to face difficulties in the final pipetting step into the 384-well plate. The Gfp fluorescence was monitored (excitation: 485, emission: 528 nM, gain: 80) every 10 min in a plate reader (Tecan Infinite 200 PRO).

The yield (objective function) in Fig. 1e, as provided in the Data availability, is the Gfp fluorescence (after 6 h incubation at 30 °C) of each composition normalized by a composition in which the concentration of all variable factors is at mid-range. However, the plotted yields are those values divided by 0.33, the average ratio of Gfp fluorescence between the active learning reference and a commonly used composition[33]. The objective function of the *LacI* circuit active learning in Fig. 3c is fold-change (FC) × dynamic range (DR) of the output (Gfp fluorescence) between 0 and 10 mM input (concentration of IPTG). For cell-free reactions in Fig. 4c, the final volume of 5 μL was prepared directly in a 384-well plate, 10 nM final concentration of each linear DNA was transferred and the mix of other components of the cell-free lysate plus T7 polymerase (40 μg.mL⁻¹) and GamS (2 μM) was added while gently mixing. The yield (objective function) in Fig. 4c is the Gfp fluorescence readout (after 6 h of incubation at 30 °C) of each transcription & translation unit normalized by the Gfp fluorescence of a commonly used sequence in our lab, wild-type T7 promoter, B0032 RBS, and *sfGfp* sequence. For all cell-free reactions, the Gfp fluorescence readout of the extract with no DNA was subtracted before yield calculations.

**RT-qPCR experiment**. Total RNA was extracted from cell-free expression reactions with a kit (NEB #T2010), following the manufacturer's instructions. Initial qPCR analysis indicated that a substantial amount of plasmid DNA remained in control reactions, which did not include reverse transcriptase to synthesize cDNA. Therefore, samples were subsequently treated to an additional DNase treatment by TURBO DNA-free™ Kit (Invitrogen™ #AM1907) according to the manufacturer's instructions. The resulting RNA produced a substantial qPCR signal (iTaq Universal SYBR Green Supermix Bio-Rad #1725120) when converted to cDNA by ProtoScript® II Reverse Transcriptase (NEB #M0368) using the standard protocol and random hexamer primers (ThermoFisher #SO142), but not in control reactions lacking reverse transcriptase. In order to account for potential sample-to-sample variability in extraction efficiency, all data presented herein is represented as a relative difference in cycle threshold (Ct) between *Gfp* and *LacI* cDNA within each sample. Standard curves with known concentrations of plasmid DNA were analyzed in parallel for *Gfp* and *LacI* primer sets, indicating comparable qPCR efficiencies and template specificity. No further normalization was required.

**Western blot**. Cell-free expression reactions and LacI-6xHis purified protein dilutions were mixed with 4 μL of non-reducing sample loading buffer (Thermo Scientific #39001) and incubated at 90 °C for 5 minutes. The samples were then loaded onto pre-cast SDS-PAGE gels (Bio-Rad #4561095) and separated by electrophoresis. The gel was then immediately placed into a Bio-Rad TransBlot® Turbo apparatus for protein transfer onto a nitrocellulose membrane (Bio-Rad #1704158). Since all samples were produced from the same batch of cell-free expression reaction mix or were of known concentration, total protein concentration was not assessed. Western blot analysis was performed using a monoclonal antibody against LacI clone 9A5 (Sigma-Aldrich #05-503-I) and an anti-mouse HRP-conjugated secondary antibody (Invitrogen #31430) with dilution of 1:1000 and 1:100,000, respectively. After dispensing the detection reagent as indicated by the manufacturer (Neogen #324175), the blot was immediately imaged on a Bio-Rad ChemiDoc. A single clear band corresponding to the molecular weight of LacI was detected in lanes containing purified LacI or expression from a *LacI*-containing plasmid (see inset, Supplementary Fig. 9a).

**In vivo experiment of transcription & translation units**. After cloning transcription & translation units into the pSEVA224 vector (plasmid and DNA preparation section), they were transformed into *E. coli* DH10β harboring an autoregulated T7 RNA polymerase circuit (addgene #71428)[64]. 3 colonies of each were cultured in LB with 30 μg.mL$^{-1}$ ampicillin + 30 μg.mL$^{-1}$ kanamycin in a 96 deep well plate. After 10 hours of cultivation at 37 °C, 10 μL of each was added to 190 μL LB with 30 μg.mL$^{-1}$ ampicillin + 30 μg.mL$^{-1}$ kanamycin in a 96-well plate (Thermo Scientific #137101). The Gfp fluorescence was monitored (excitation: 485, emission: 528 nM, gain: 80) every 30 min in a plate reader (Tecan Infinite 200 PRO) shaking at 37 °C. The in vivo yield in Fig. 4e, f is the Gfp fluorescence readout (after 6 h) of each transcription & translation unit normalized by the Gfp fluorescence of a commonly used sequence in our lab, wild-type T7 promoter, B0032 RBS, and *sfGfp* sequence. The Gfp fluorescence readout of cells with no *sfGfp* gene was subtracted before yield calculations.

**Workflow for CETCH assays in 384-well plates**. The worklist generated by the METIS script was dissected into 5 worklists: dH$_2$O, Buffers and Cofactors, Enzymes, Carbonic Anhydrase, and Substrate (pco: propionyl-CoA oxidase, ccr: crotonyl-CoA carboxylase/reductase, epi: ethylmalonyl-CoA/methylmalonyl-CoA epimerase, mcm: methylmalonyl-CoA mutase, scr: succinyl-CoA reductase, ssr: succinic semialdehyde reductase, hbs: 4-hydroxybutyryl-CoA synthetase, hbd: 4-hydroxybutyryl-CoA dehydratase, ecm: ethylmalonyl-CoA mutase, mco: methylsuccinyl-CoA oxidase, mch: mesaconyl-CoA hydratase, mcl: β-methylmalyl-CoA lyase, gor: glyoxylate reductase, kat: catalase, fdh: formate dehydrogenase, ck: creatine phosphokinase). For source of enzymes and kinetic parameters see Schwander et al.[39]. In cases where pipetting errors occurred, we used our Exceptions_to_Worklist script for correction of failed transfers (provided in Code availability). This script generates a new worklist out of the exception file generated by the ECHO® and provides a list with how much volume needs to be added into which well. Dissecting the worklists guarantees for example that all buffers are transferred before enzymes are added. Note that we used fresh enzyme stocks in each round to prevent loss of activity due to repetitive freeze-thaw cycles. As source plates we used ECHO® qualified 384-Well PP 2.0 Plus Microplates from Labcyte and used AQ_GP as the liquid class (AQueous solution; Glycerol/Protein). This liquid class was tested previously with the stocks of our assay components.

We also added a control condition with composition derived from the published assay of CETCH 5.4 (for the composition, see Assays for determination of new enzyme stocks after round two section in Supplementary Note 4). Controls can be added in the workflow as specials. The yield of our control condition increased from round 2 to 3, where new enzyme batches of four enzymes were used (Supplementary Fig. 16b). To identify the enzyme that was the reason for that, we tested the control assay with each of the four old enzymes separately (Supplementary Fig. 16b). Despite being important in the control (~280 μM in round 1 and 2), catalase did not seem important in each condition, since we reached yields up to 1500 μM already in round 2 with the old stock (Fig. 5c).

After starting the assays with 100 μM propionyl-CoA we used an Axygen® Breathable Sealing Film (BF-400-S) to cover the 384-well PCR Plate (AB-1384) to allow the transfer of oxygen. The reaction (10 μL volume) was carried out at 30 °C and mild shaking at 160 rpm in an Infors HT Ecotron shaker. The reactions were stopped after 3 h with 1.25 μL of 500 mM polyphosphate and 1.25 μL of 50% formic acid. While the formic acid quenches the reaction, the polyphosphate was used for enhanced precipitation of the proteins. The plate was spun for 1 h at 2272 g and 4 °C to pellet the proteins.

For analysis by LC-MS, we used a multichannel pipette to transfer 1 μL of the supernatant into 9 μL of precooled dH$_2$O in a new 384-Well Thermo-Fast® plate. Afterward, we added 10 μL of 10 μM $^{13}$C$_2$ labeled glycolic acid as an internal standard. The plate was sealed with a Corning™ Microplate Aluminum Sealing Tape (6570). The assay plate with the quenched reactions was sealed with a Corning™ Microplate Aluminum Sealing Tape too and stored at −80 °C.

**Timepoint assays of 7 selected conditions**. The assays were done in triplicates containing 150 μL volume each and were carried out in a 1.5 mL reaction tube (at 30 °C, 500 rpm). The reactions were started with 100 μM propionyl-CoA. 12 μL samples were taken and quenched with 1.5 μL 50% formic acid and 1.5 μL 500 mM sodium polyphosphate (emplura®) at 5, 10, 15, 30, 60, 120, 180, 240, 300 and 480 min. The samples were spun for 20 min at 4 °C and 20.000 g, before the supernatant was transferred into Thermo Scientific™ Abgene 96 Well Polypropylene Storage Microplates (AB-1058) and sealed with Corning™ Microplate Aluminum Sealing Tape. While 2 μL were used to prepare a 1:10 dilution in water for the measurement via LC-MS, the remaining samples were stored at −80 °C. The concentrations for the assays are shown in the table below (Buffers and cofactors in mM, enzymes in μM). See Supplementary Table 3 for the details of these conditions.

**LC-MS analysis of CoA esters**. All CoA esters were measured on a triple quadrupole mass spectrometer (Agilent Technologies 6495 Triple Quad LC-MS) equipped with a UHPLC (Agilent Technologies 1290 Infinity II) using a 150 × 2.1 mm C18 column (Kinetex 1.7 μm EVO C18 100 Å) at 25 °C. The injection volume was 2 μL of the diluted samples (1:10 in water). The flow was set to 0.400 mL.min$^{-1}$ and the separation was performed using 50 mM ammonium

formate pH 8.1 (buffer A) and acetonitrile (buffer B). We quantified the CoAs using external standard curves prepared in water with formic acid at pH 3. The standard curves were measured before and after the samples. Except for methylsuccinyl-CoA, all compounds were stable. For methysuccinyl-CoA we calculated the concentration as an average of the two standard curves at the time point the sample was measured. The parameters for the multiple reaction monitoring (MRMs) and the gradient are shown in the tables below. The data analysis was done with Agilent MassHunter Quantitative Analysis (for QQQ). See Supplementary Table 4 (Gradient for the separation of CoA esters) and Supplementary Table 5 (MRM transitions).

**LC-MS analysis of glycolate**. Glycolate was measured on a triple quadrupole mass spectrometer (Agilent Technologies 6495 Triple Quad LC-MS) equipped with a UHPLC (Agilent Technologies 1290 Infinity II) using a 150 ×2.1 mm C18 column (Kinetex 1.7 μm EVO C18 100 Å) at 25 °C. The injection volume was 0.5 μL. The diluted samples (1:10 in water), as well as the external standard curve, were diluted 1:2 with 10 μM $^{13}$C$_2$-labeled glycolic acid as internal standard. The flow was set to 0.100 mL.min$^{-1}$ and the separation was performed using dH$_2$O with 0.1% formic acid (buffer A) and methanol with 0.1% formic acid (buffer B). The parameters for the multiple reaction monitoring (MRMs) and the gradient are displayed below. Data analysis was done using the Agilent Mass Hunter Workstation Software. See Supplementary Table 6 (Gradient for the separation of CoA esters) and Supplementary Table 7 (MRM transitions).

**Data analysis**. Data was analyzed using Microsoft Excel, GraphPad Prism, and custom Python scripts (available at https://github.com/amirpandi/METIS) and Agilent Mass Hunter Workstation Software (QQQ) 10.0 for LC-MS data.

**Reporting summary**. Further information on research design is available in the Nature Research Reporting Summary linked to this article.

## Data availability
The source data underlying Figs. 3f-i, 4a, e, and 5h-l and Supplementary Figures 2, 7a, b, 9b, 11, 16a, b are provided as a Source Data file. All active learning data (combinations and yields) are available along with Google Colab Python notebook of each application on GitHub, https://github.com/amirpandi/METIS. Primers used for transcription and translation units (Fig. 4) are provided in Supplementary Tables 8, 9. Source data are provided with this paper.

## Code availability
METIS workflows for different applications used in this study run as Google Colab Python notebooks and are free open source tools available at https://github.com/amirpandi/METIS. All scripts used in this study were written in Python 3. Packages used in the development of the workflow were pandas (1.1.4) and numpy (1.18.5), matplotlib (3.2.2) and seaborn (0.11.0), scikit-learn (0.22.2.post1), xgboost (0.90), and Keras (2.3.1) using TensorFlow backend.

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

## Acknowledgements

We wish to thank V. Gureghian for stimulating discussions during conception of the work, T. E. Gorochowski (University of Bristol, UK) for providing SLC and MLC constructs, and M. Kushwaha (INRAE, Jouy en Josas, France) for *E. coli* DH10β, harboring the auto-regulated T7 polymerase construct. We thank S. Burgener, S. Luo, A. Sánchez-Pascuala Jerez, N. Odermatt, and A. Schrodt for graphical, technical and experimental support, as well as fruitful discussion. We thank Jule P. M. Erb for drafting the METIS logo. This work was supported by a European Molecular Biology Organization (EMBO) long-term post-doctoral fellowship (A.P. ALTG 165-2020), the Gordon and Betty Moore Foundation, GBMF10652, grant DOI 10.37807/GBMF10652 (T.J.E.), the Max Planck Research Network in Synthetic Biology (MaxSynBio) of the Max Planck Society and the Federal Ministry of Education and Research (BMBF; T.J.E.), the UDOPIA PhD program and the French National Research Institute for Agriculture, Food, and Environment (INRAE) through the Métaprogramme BIOLPREDICT program (L.F.), BMBF Grant, MetAFor, No. 031B0850B (T.J.E.), ANR iCFree grant (ANR-20-BiopNSE) (J-L.F.) and the Max-Planck Society (MPG-FhG project eBioCO₂n). Figures were created with Biorender.com.

## Author contributions

T.J.E. and A.P. conceived the work. A.P. and C.D. designed the workflow and performed experiments. A.Y.K. wrote all Python notebooks and simulations. L.F. and J-L.F. reviewed the workflow's code and consistency and provided constructive inputs. S.A.S. performed western blot and RT-qPCR experiments. E.B. implemented the DART algorithm and SMOGN data preprocessing. M.N. prepared the enzyme mutants dataset. D.A, N.C., Y.F., and C.M. assisted with the experimental work. N.P. developed the LC-MS methods for the analysis of glycolate and CoA esters. N.S.C. wrote the code for the

conversion of the ECHO® exceptions lists into new worklists. T.J.E. supervised the work. A.P., C.D., A.Y.K. and T.J.E. wrote the manuscript with input from all other authors.

## Funding

## Competing interests
The authors declare no competing interests.
