## [Peer Review File · Nature Communications]

Reviewers' Comments:

Reviewer #1:

Remarks to the Author:

Summary:

In this work, the authors developed a machine learning platform with simplified software called METIS (Machine-learning guided Experimental Trials for Improvement of Systems) and applied this workflow to optimize increasingly complex systems with informed experimentation to reduce the test space. The formulation of a cell-free protein synthesis (CFPS) reaction was optimized 30-fold, the output of a linear expression template CFPS was improved 12-fold, the productivity and efficiency of the synthetic carbon fixation cycle (CETCH) were improved 10- and 6-fold, respectively, and a cell-free genetic circuit was optimized as well. Through the development of the METIS platform, the authors sought to make active learning more accessible to experimentalists by simplifying the coding requirements and providing straightforward outputs that could be implemented by manual pipetting or acoustic liquid handling.

Conceptual novelty, scientific accuracy, and repeatability:

Overall, this work went further than comparable active learning papers in showing diverse applications and significant optimizations. The experimental novelty is low as the authors built on existing studies to develop and validate their workflow, but the workflow itself and the breadth of applications combine to make this a novel effort. The authors attempt to make machine learning more accessible to experimentalists and demonstrate this over an impressive range of experiments with unique outputs to achieve significant improvements across all systems tested. The extensive detail provided in the supplementary notes should enable reproducibility and broader use of the METIS platform. However, it is unclear how easily an experimentalist with minimal coding expertise would be able to adapt the parameters to different systems. Otherwise, the claims are well supported by references and/or data. The CETCH cycle optimization and validation is particularly thorough, providing a convincing final application. The article is well written and this paper is on the leading edge of innovation for synthetic biology.

Major technical criticisms/questions:

- The choice of active learning algorithm is not sufficiently explained. MLP and DNN are known to require larger sample sizes, as the authors state on page 2 lines 22-23. The performance of linear regressors and XGBoost appear comparable in the top portion of Figure 1b, and the linear regression plot appears to have a tighter distribution (aside from outliers) from Round 4 on. Why was linear regression discarded? The bottom portion of Figure 1b then shows another panel of XGBoost optimization with 100 samples per round that has a tighter distribution. Was this simply the result of stochasticity in the active learning algorithm?
- The authors claim to reach 30x higher GFP yields than a commercial kit (page 3, line 1), but they only produce ~15 μM (400 $\mu\text{g}/\text{ml}$) compared to "myTXTL toolkit 2.0" (<https://doi.org/10.1021/acssynbio.5b00296>) that produced >2 mg/ml GFP using the same base reaction formulation (although the cell lysis methods differed). The dataset from Borkowski et al. that this optimization was based on appears to use an inferior plasmid (or less clean plasmid preparation) as they were able to produce much higher GFP concentrations using the plasmid provided in the commercial kit (<https://doi.org/10.1038/s41467-020-15798-5>, supplementary figure 8). This difference in baseline productivity makes the results of reaction optimization questionable, as the optimum formulation might not be transferrable to a more active system (i.e., the optimization here addressed different limitations than would be present in reactions expressing 5x as much protein in the same timespan). Do your optimized reaction conditions also increase protein synthesis when this higher yielding plasmid is used? This should be tested. What about moving beyond these yields to optimize the well-tuned and already high yielding cell-free protein expression systems? There are several reports of > 2.5 mg/ml in batch.
- The section about the LacI sensor optimization reads a bit odd. The defined problem, "we aimed at using our workflow to increase both the dynamic range and fold-change of in vitro protein production for the SLC and MLC circuits" does not seem to be fully addressed. Instead, the section uncovers the introduced problem of resource competition when adding the pT7-LacI plasmid to the cell free sensing reaction along with the GFP reporter. This is fixed for the SLCs (Figure 3K) by

exogenously adding purified LacI but is not tested for MLCs.

- The initial CETCH cycle ran for 90 minutes, while the assays reported here last for 8 hours. Was additional optimization performed between these papers to extend the reaction?

Minor technical criticisms/questions:

- Page 1 Line 8 and Page 15 Line 39. "intense" should be "intensive"
- Page 1 Line 46 and 47. "So far" was used twice in the sentence.
- Page 2 Line 13 and 14. Check the phrase 'gold regressor'?
- Page 3 Line 2. Supplementary Fig. 1 does not do a good job of relaying the claims in the text. Please make a figure of the comparison you are trying to make. The figure, currently, is a standard curve and gives no information to the comparison of their optimized TXTL with previously optimized systems.
- Error bars are shown for the "% Feature Importance" of LET architecture in Supplementary Figure 10b, but no error is shown for the other optimizations in which the Feature Importance output is included in the main text.
- Optimizing genetic circuits is more complex than the LET optimization shown afterward. Is there a reason you presented them in this order?
- Page 11 line 5 states that the CETCH cycle has 26 components, while other instances indicate 27 (in the abstract, Figure 5, and discussion).
- SI Pg 26. There is a typo in Supp Fig 18a. "Yild"

Reviewer #2:

Remarks to the Author:

The authors present an active learning framework to optimize biological systems with minimal experimentation. The work nicely combines machine learning with experimentation and demonstrates the approach on multiple diverse biological systems. They develop an online tool so other researchers can implement active learning on their own systems. While this work is thorough and the developed tools are generally useful, I feel the presentation is not sufficiently detailed and clear to be published at this point. Some specific comments:

1. The approach (UCB) the authors describe is not typically considered "active learning", but instead Bayesian optimization or bandit optimization. These concepts are closely related. The goal of active learning is to efficiently learn the entire functional landscape by selecting informative points, while Bayesian optimization seeks to identify the best points from the landscape by trading off exploration and exploitation. The authors may consider revising their terminology to be consistent with the field.
2. The machine/active learning methods were not described in sufficient detail. How was UCB for linear regression calculated? Did they use analytical prediction intervals or the (deterministic) ensemble approach? There were no details on MLP and DNN. How many layers? Convolutional, recurrent, transformers?
3. During each round, experiments were performed in batches. I didn't understand how these batches were selected. Did they generate random combinations of experiments and choose the batch with the highest average UCB? Was there any effort to diversity the designs within each batch? See <https://www.pnas.org/doi/10.1073/pnas.1215251110> for example.
4. Looking at Fig1b, it seems linear regression significantly outperforms XGBoost. It learns faster, discovers high yield combinations efficiently, and has tighter distributions. Why was XGBoost chosen over linear regression?
5. All the examples given involved optimizing in vitro systems over input concentrations. Could the same approaches be used to optimize over genetic components (promoters, genes, pathways) within cells?

We thank the reviewers for their valuable comments that provided us with the opportunity to improve our work further. Please find below a point-by-point **responses** and **actions** to all comments of the reviewers.

Reviewer #1 (Remarks to the Author):

Summary:

In this work, the authors developed a machine learning platform with simplified software called METIS (Machine-learning guided Experimental Trials for Improvement of Systems) and applied this workflow to optimize increasingly complex systems with informed experimentation to reduce the test space. The formulation of a cell-free protein synthesis (CFPS) reaction was optimized 30-fold, the output of a linear expression template CFPS was improved 12-fold, the productivity and efficiency of the synthetic carbon fixation cycle (CETCH) were improved 10- and 6-fold, respectively, and a cell-free genetic circuit was optimized as well. Through the development of the METIS platform, the authors sought to make active learning more accessible to experimentalists by simplifying the coding requirements and providing straightforward outputs that could be implemented by manual pipetting or acoustic liquid handling.

Conceptual novelty, scientific accuracy, and repeatability:

Overall, this work went further than comparable active learning papers in showing diverse applications and significant optimizations. The experimental novelty is low as the authors built on existing studies to develop and validate their workflow, but the workflow itself and the breadth of applications combine to make this a novel effort. The authors attempt to make machine learning more accessible to experimentalists and demonstrate this over an impressive range of experiments with unique outputs to achieve significant improvements across all systems tested. The extensive detail provided in the supplementary notes should enable reproducibility and broader use of the METIS platform. However, it is unclear how easily an experimentalist with minimal coding expertise would be able to adapt the parameters to different systems. Otherwise, the claims are well supported by references and/or data. The CETCH cycle optimization and validation is particularly thorough, providing a convincing final application. The article is well written and this paper is on the leading edge of innovation for synthetic biology.

Response:

We thank the reviewer for his/her very positive evaluation of our study and the thoughtful comments and questions, which we addressed point-by-point below.

Regarding the ease application of METIS for experimentalists with minimal programming skills, we thank the reviewer for acknowledging our efforts to provide a detailed description of workflows in **Supplementary Note 2** and **Supplementary Fig. 4-6**, as well as an extensive organization of modules in the Google Colab notebooks. To exemplify different applications, we decided to showcase diverse examples in our manuscript and plan to provide additional online resources (in form of tutorials) in the future. On a side note, we tested METIS in our own lab and made very good experience with different users from non-bioinformatics background. This provides us with hope that the workflow will find its broader application in the community. We also hope that a prominent publication will further promote the use of METIS.

Major technical criticisms/questions:

- The choice of active learning algorithm is not sufficiently explained. MLP and DNN are known to require larger sample sizes, as the authors state on page 2 lines 22-23. The performance of linear regressors and XGBoost appear comparable in the top portion of Figure 1b, and the linear regression plot appears to have a tighter distribution (aside from outliers) from Round 4 on. Why was linear regression discarded?

Response:

Thank you for providing us with the opportunity to explain our choice of workflow better. In **Fig. 1b**, we show why we selected XGBoost over other models for METIS. As the reviewer already pointed out, MLP and DNN models need much larger datasets for training although they are able to fit virtually any function (Universal Approximation). Because of this data inefficiency, we discarded MLP and DNN models. With small datasets, the linear regression model was not able to achieve a similar performance in our simulations compared to XGBoost (please see **Supplementary Fig. 1** in the revised version). Therefore, we decided to use XGBoost, which is also known as a powerful model for capturing non-linear interaction, and as a versatile algorithm that can be applied on different systems (ref. 40).

Tight or wide distribution of data points is because of stochasticity, although the general behavior of the active learning remains the same. For example, in **Supplementary Fig. 1** the XGBoost plot (100 data points) have tighter distribution than linear regression.

Action:

We now explicitly mention the difference between XGBoost and linear regressor in the main text and explain the choice for XGBoost in more detail on P2L26-33:

*“XGBoost outperformed linear regressors when fewer data points per round were used (**Supplementary Fig. 1**).*

XGBoost is an improved random forest-type algorithm, working through gradient boosted decision trees⁴⁰ by aggregating and compiling sets of models. This algorithm is a sparsity-aware, fast, scalable as well as versatile model for handling tabular data with complex non-linear interactions⁴⁰. These features make XGBoost a promising algorithm for machine learning applications on different biological systems with limited datasets. For our workflow, we therefore selected XGBoost, which has also been used for different biological applications previously^{18,41,42}.”

In the revised version of the manuscript, we also implemented an optional feature of XGBoost package called “DART” that improves the model performance through reducing overfitting (**Supplementary Fig. 21a**).

The bottom portion of Figure 1b then shows another panel of XGBoost optimization with 100 samples per round that has a tighter distribution. Was this simply the result of stochasticity in the active learning algorithm?

Response:

The reviewer is correct. This is because of both stochasticity of the model and randomness in selecting data points.

- The authors claim to reach 30x higher GFP yields than a commercial kit (page 3, line 1), but they only produce ~15 μM (400 $\mu\text{g/ml}$) compared to “myTXTL toolkit 2.0” (<https://doi.org/10.1021/acssynbio.5b00296>) that produced >2 mg/ml GFP using the same base reaction formulation (although the cell lysis methods differed). The dataset from Borkowski et al. that this optimization was based on appears to use an inferior plasmid (or less clean plasmid preparation) as they were able to produce much higher GFP concentrations using the plasmid provided in the commercial kit (<https://doi.org/10.1038/s41467-020-15798-5>, supplementary figure 8). This difference in baseline productivity makes the results of reaction optimization questionable, as the optimum formulation might not be transferrable to a more active system (i.e., the optimization here addressed different limitations than would be present in reactions expressing 5x as much protein in the same timespan). Do your optimized reaction conditions also increase protein synthesis when this higher yielding plasmid is used? This should be tested. What about moving beyond these yields to optimize the well-tuned and already high yielding cell-free protein expression systems? There are several reports of > 2.5 mg/ml in batch.

Response:

We apologize for the misunderstanding. Our target protein not GFP, but was super-folder GFP (sfGFP). Note that the yield of sfGFP was reported to be much lower compared to GFP (and other proteins) in previous publications (please refer to the supplementary information of Borkowski et al. **Figure 8b**). We understand that this might cause some confusion. Thus, we removed the claim from the main text and exclusively focus on the 20-fold improvement compared to the control setup.

Action:

We removed the comparison statement in **Supplementary Fig. 1** (of the previous version of our manuscript that is **Supplementary Fig. 2** in the revised version) and the main text.

- The section about the LacI sensor optimization reads a bit odd. The defined problem, “we aimed at using our workflow to increase both the dynamic range and fold-change of in vitro protein production for the SLC and MLC circuits” does not seem to be fully addressed. Instead, the section uncovers the introduced problem of resource competition when adding the pT7-LacI plasmid to the cell free sensing reaction along with the GFP reporter. This is fixed for the SLCs (Figure 3K) by exogenously adding purified LacI but is not tested for MLCs.

Response:

Thank you for allowing us to clarify our experimental strategy better.

- 1) DYNAMIC RANGE AND FOLD CHANGE: The reason we chose both dynamic range and fold-change is because considering only fold-change as objective function may lead to scoring combinations with high fold-change but very low total protein production. As shown in **Supplementary Fig. 7**, while initial pTAC and pTHS circuits show the same fold-change, pTAC results in a very low protein production. In other words, with a very

low OFF state (i.e., protein production in the absence of IPTG), even low changes in protein production might result in a high fold-change. To avoid being trapped in such combinations, we also took the dynamic range into account, which is the difference in protein production between OFF and ON states, with 0 and 10 mM of IPTG respectively. By setting “dynamic range multiply by fold-change” as the objective function, we scored those combinations favoring the multiplication product of both features. We re-wrote the section to emphasize this point better (see **action** below).

- 2) **SLC AND MLC CIRCUITS:** Regarding SLC and MLC circuits, all four circuits (pTAC, pTHS, pSTAR, pDC) were alternatively used as a factor in active learning. pTHS was the best performing circuit under optimal conditions (**Fig. 3d** and **Supplementary Note 3**). We added some more explanation to the text to explain this better (see **action** below).
- 3) **RESOURCE COMPETITION:** We initially addressed the resource competition question with the experiment in **Fig. 3f** on pTHS (optimal condition from active learning), which is an MLC construct. Increasing *LacI* plasmid concentrations did negatively affect function of the circuit. We then switched back to a simple system of a *GFP*-expressing and a *LacI*-expressing plasmid, respectively (**Fig. 3g**) to demonstrate the resource competition effect. Finally, we verified the resource competition scenario at the protein (**Fig. 3h**) and mRNA (**Fig. 3i**) levels, demonstrating that our *LacI* circuits could not be substantially improved due to resource limitation. After having replaced the *LacI* plasmid with purified *LacI* protein, we used 20 most informative combinations in the workflow. Note that among these 20 combinations were again all four SLC and MLC circuits. All of them improved upon providing external *LacI*, clearly demonstrating that resource competition had been limiting performance of the SLC, as well as the MLC circuits. All combinations and their associated yields are available on the METIS GitHub repository.

Action:

To clarify these points above, we made the following modifications on P7L11-33:

*“Additional to the high fold-change (FC), a desired circuit should have a high level of protein production, a feature that can be quantified by the dynamic range (DR) (**Fig. 3a**).*

*Here, we aimed at using our workflow to optimize the SLC and MLC *LacI* circuits. We performed 10 rounds of active learning with the objective function of $FC \times DR$, to score those compositions that result in not only high fold-changes but also high total *Gfp* productions. The fold-change can be improved by supplying an additional plasmid expressing *LacI* under the control of a T7 promoter (transcribed by purified T7 RNA polymerase) and the dynamic range can be improved through alternative selection of SLC and MLC circuits and through tuning TXTL composition. The active learning cycle received input from several factors in the *E. coli* cell-free system; amino acids and tRNAs, which are important when extra DNA is added, DTT as reducing reagent, spermidine for DNA-protein binding, and PEG 8000 as crowding agent (**Fig. 3b**), and four *LacI* circuits (one SLC and three MLC) were considered as one categorical feature with four alternatives. While the objective function improved during the active learning cycle (bottom plot in **Fig. 3c**), we did not observe a substantial improvement in fold-change of *Gfp* production alone (upper plot in **Fig. 3c**). Feature importance analysis identified the concentration of the P_{T7} -*LacI* plasmid as strong contributor (**Fig. 3d**, **Fig. 3e**, **Supplementary Fig. 8**), indicating deleterious*

Lacl-protein/DNA interactions or resource limitation of the TXTL system through production of the lacl protein⁴⁴.

*By performing a titration experiment with P_{T7} -Lacl, we showed that addition of the Lacl plasmid has indeed a strong negative effect on the optimal Lacl circuit (i.e., with pTHS) (Fig. 3f) and in an independent TXTL protein production (Fig. 3g) (see also **Supplementary Note 3** for details of the active learning cycle and titration experiments)."*

P8L2-4: "Note that among these 20 combinations were again all four SLC and MLC circuits. All of them improved upon providing external Lacl, clearly demonstrating that resource competition had been limiting performance of the SLC, as well as the MLC circuits."

- The initial CETCH cycle ran for 90 minutes, while the assays reported here last for 8 hours. Was additional optimization performed between these papers to extend the reaction?

Response:

Thank you for asking. We did not use initial CETCH cycle setup (Schwander et al. Science 2016), but a setup recently published by us (Miller et al. Science 2020) and used a different ATP regeneration system, as higher concentrations of the initial regeneration substrate (polyphosphate) cause precipitation at higher concentrations. While we indeed run the assay over 8 hours, most of the product is formed in the first 90-120 minutes, as shown in **Fig. 5h**.

Minor technical criticisms/questions:

- Page 1 Line 8 and Page 15 Line 39. "intense" should be "intensive"

Response:

We thank the reviewer for pointing this out. We corrected the typo (P1L8).

- Page 1 Line 46 and 47. "So far" was used twice in the sentence.

Response:

We thank the reviewer for pointing this out. We corrected the typo (P1L47).

- Page 2 Line 13 and 14. Check the phrase 'gold regressor'?

Response:

We added a brief explanation about the "gold regressor"

Action: P2L15:

"...a gold regressor (a reference model fitted on pre-existing experimental data to evaluate new algorithms)..."

- Page 3 Line 2. Supplementary Fig. 1 does not do a good job of relaying the claims in the text. Please make a figure of the comparison you are trying to make. The figure, currently, is a

standard curve and gives no information to the comparison of their optimized TXTL with previously optimized systems.

Response:

We agree with the reviewer. The aim of experiment shown in **Supplementary Fig. 1** (now **Supplementary Fig. 2** in the revised version) was to provide a standard curve for GFP readout, not to provide a comparison with other setups. Since this seems to be misleading and distracting the reader, we removed the comparison claims.

Action:

We removed the comparison statement in the **Supplementary Fig. 1** (now **Supplementary Fig. 2** in the revised version) and the main text.

- Error bars are shown for the “% Feature Importance” of LET architecture in Supplementary Figure 10b, but no error is shown for the other optimizations in which the Feature Importance output is included in the main text.

Response:

We thank the reviewer for pointing out this apparent mistake in calculating feature importance. Feature importance algorithm does not provide an error bar and the previous versions of the plot had mean and std of importance from all rounds. The feature importance is calculated cumulatively taking into account data from previous rounds, hence we do not need to take mean and std of all rounds into account.

Action:

In the revised version, we corrected **Supplementary Fig10b** and removed all error bars (and cumulative feature importance values plotted for the latest round)

- Optimizing genetic circuits is more complex than the LET optimization shown afterward. Is there a reason you presented them in this order?

Response:

The reviewer is right that genetic circuit optimization is more complex than LET optimization. However, due to the biological logic within the manuscript, we decided to present the optimization of the cell-free system first, before that of the genetic circuit, which is built on an optimized LET.

- Page 11 line 5 states that the CETCH cycle has 26 components, while other instances indicate 27 (in the abstract, Figure 5, and discussion).

Response:

We corrected to “27” (P11L32).

- SI Pg 26. There is a typo in Supp Fig 18a. “Yild”

Response:

We corrected to “yield”.

Reviewer #2 (Remarks to the Author):

The authors present an active learning framework to optimize biological systems with minimal experimentation. The work nicely combines machine learning with experimentation and demonstrates the approach on multiple diverse biological systems. They develop an online tool so other researchers can implement active learning on their own systems. While this work is thorough and the developed tools are generally useful, I feel the presentation is not sufficiently detailed and clear to be published at this point. Some specific comments:

Response:

We are grateful to the reviewer for the overall positive comments and have carefully revised our manuscript, as suggested by the reviewer to increase the presentation of our work.

1. The approach (UCB) the authors describe is not typically considered “active learning”, but instead Bayesian optimization or bandit optimization. These concepts are closely related. The goal of active learning is to efficiently learn the entire functional landscape by selecting informative points, while Bayesian optimization seeks to identify the best points from the landscape by trading off exploration and exploitation. The authors may consider revising their terminology to be consistent with the field.

Response:

We agree with the reviewer that the technical fully correct term is “Bayesian optimization”, which we had mentioned in the methods section before (P18L11), but now also mention explicitly in the introduction and discussion (see **action**).

However, we would prefer to keep “active learning” as colloquial phrase throughout the text, as it has been previously used in the field of synthetic biology in the same context and also might be better understandable for experimentalists.

Action:

To further clarify the definition of the algorithms used, we added the following statements to the introduction and discussion:

Introduction (P1L19-20): “Note that, active learning for optimizing a system is also known as Bayesian optimization”

Discussion (P15L32): “We provide Google Colab notebooks (see Code availability) that can be adapted to different optimization applications (also known as Bayesian optimization) and even used for data-driven predictions”

Supplementary Table 1: “Summary of METIS optimization (also known as Bayesian optimization) and prediction Google Colab packages.”

2. The machine/active learning methods were not described in sufficient detail. How was UCB for linear regression calculated? Did they use analytical prediction intervals or the (deterministic) ensemble approach? There were no details on MLP and DNN. How many layers? Convolutional, recurrent, transformers?

Response:

We thank the reviewer for these important questions. We updated the Methods section to provide more detail on UCB calculation and machine learning models architecture. In case of UCB calculation for the linear model, given its deterministic nature, the UCB was calculated only by the predicted value (std equal to zero). We have considered Gaussian Process (GP) regression that comes with uncertainty for each predicted point. However, since the optimal kernel can be highly variable between different systems, we decided not to use GP for METIS, which aimed to be a more versatile, pragmatic tool.

Action:

We amended the text by the following statements (P17L19-26):

“Hyperparameters: MLPRegressor from Sklearn (fully connected architecture with Relu activation function) was used for MLP. In ensemble of 5 models the following number of neurons were used in the hidden layer: (10, 100, 100, 20), (20, 100, 100, 10), (20, 100, 100, 20), (10, 100, 100, 10), (20, 100, 100, 50). For DNN we used the Keras implementation of fully connected layer architecture with 100, 100, 20 neurons for each of hidden layers. For Linear Regression the default implementation of Ordinary Least Square by Sklearn was used. XGBRegressor with following parameter was used for XGBoost model: objective = 'reg:squarederror', n_estimators = 500, learning_rate = 0.01, max_depth = 6, min_child_weight = 1, subsample = 0.8.”

3. During each round, experiments were performed in batches. I didn't understand how these batches were selected. Did they generate random combinations of experiments and choose the batch with the highest average UCB? Was there any effort to diversity the designs within each batch? See <https://www.pnas.org/doi/10.1073/pnas.1215251110> for example.

Response:

In each round, the yield for thousands of random combinations were predicted by the ensemble of models. Using the calculated UCB, we sorted these combinations and selected the top combinations for the next round, which is explained in the **Methods** section. To avoid getting trapped at local maxima, especially in the beginning, we tuned the “exploration” coefficient in each round. We typically started with a high exploration and decreased the exploration coefficient towards later rounds. The exploration of models and the exploration strategy, as well as a guide of how to define parameters during different rounds of active learning (and also for different biological systems) are discussed in more detail in **Supplementary Note 2**.

4. Looking at Fig1b, it seems linear regression significantly outperforms XGBoost. It learns faster, discovers high yield combinations efficiently, and has tighter distributions. Why was XGBoost chosen over linear regression?

Response:

Thank you for providing us with the opportunity to explain our choice of workflow better. In **Fig. 1b** we show why we selected XGBoost over other models for METIS. As the reviewer already pointed out, MLP and DNN models need much larger datasets for training although they are able to fit virtually any function (Universal Approximation). Because of this data inefficiency, we discarded MLP and DNN models. With small datasets, the linear regression model was not able

to achieve a similar performance in our simulations compared to XGBoost (please see **Supplementary Fig. 1** in the revised version). Therefore, we decided use XGBoost, which is also known as a powerful model for capturing non-linear interaction, and as a versatile algorithm that can be applied on different systems (ref. 40).

Action:

We now explicitly mention the difference between XGBoost and linear regressor in the main text and explain the choice for XGBoost in more detail on P2L24-31:

*“XGBoost outperformed linear regressors when fewer data points per round were used (**Supplementary Fig. 1**).*

XGBoost is an improved random forest-type algorithm, working through gradient boosted decision trees⁴⁰ by aggregating and compiling sets of models. This algorithm is a sparsity-aware, fast, scalable as well as versatile model for handling tabular data with complex non-linear interactions⁴⁰. These features make XGBoost a promising algorithm for machine learning applications on different biological systems with limited datasets. For our workflow, we therefore selected XGBoost, which has also been used for different biological applications previously^{18,41,42}.”

In the revised version of the manuscript, we also implemented an optional feature of XGBoost package called “DART” that improves the model performance through reducing overfitting (**Supplementary Fig. 21a**).

5. All the examples given involved optimizing *in vitro* systems over input concentrations. Could the same approaches be used to optimize over genetic components (promoters, genes, pathways) within cells?

Response:

Indeed, any measurable objective (target) function depending on categorical and/or continuous variables can be optimized using METIS. We demonstrated this for systems with categorical variables *e.g.*, promoter and RBS type (**Fig. 4**) and continuous variables *e.g.*, salts and proteins concentration. As long as a user can change (and/or experimentally control) variables (*i.e.*, values or categories), there is no fundamental difference between *in vivo* and *in vitro* systems, because dependency of the measurable output on variables and interactions among variables can be captured by METIS. These features of our workflow and how it can be applied also for *in vivo* systems (regulatory elements, constructs, and metabolic pathways) are discussed in **Supplementary Note 2**.

Reviewers' Comments:

Reviewer #1:

Remarks to the Author:

Pandi et al. have sufficiently addressed the concerns of the reviewers. Optimizing genetic and metabolic networks is of great importance to the field, and this work makes machine learning driven experimentation accessible to the average user. We believe METIS will be immediately implemented by others and will have a significant impact on the field at large. We recommend publication as is with one last minor change to the supplemental.

Supplemental P3L149: "Results.scv" is probably supposed to be "Results.csv"

Reviewer #2:

Remarks to the Author:

The authors have mostly addressed my comments. I now understand the approach better based on their response to my initial comments.

I do see one limitation related to choosing experiments in batches that could warrant further discussion. The approach samples thousands of random combinations, evaluates the UCB score for each, and then sorts and takes the top. The UCB score is calculated independently for each sample and therefore this approach will have the tendency to converge to similar solutions with high UCB scores. This results in low diversity experiments that provide limited new information.

There is a vast amount of literature on this batch UCB setting, e.g.:

<https://proceedings.mlr.press/v70/daxberger17a.html> . Incorporating batch UCB approaches would likely improve the METIS method because each design-test-learn cycle would generate richer data for subsequent rounds.

We thank the reviewers for their feedback and suggestions that provided us with the opportunity to improve our work. Please find below a point-by-point **responses** and **actions** to all comments of the reviewers.

Reviewer #1 (Remarks to the Author):

Pandi et al. have sufficiently addressed the concerns of the reviewers. Optimizing genetic and metabolic networks is of great importance to the field, and this work makes machine learning driven experimentation accessible to the average user. We believe METIS will be immediately implemented by others and will have a significant impact on the field at large. We recommend publication as is with one last minor change to the supplemental.

Response:

We are glad that the reviewer found our responses/actions satisfactory addressing her/his comments. We again thank the reviewer for the positive feedback.

Supplemental P3L149: "Results.scv" is probably supposed to be "Results.csv"

Response:

We corrected to "Results.csv".

Reviewer #2 (Remarks to the Author):

The authors have mostly addressed my comments. I now understand the approach better based on their response to my initial comments.

I do see one limitation related to choosing experiments in batches that could warrant further discussion. The approach samples thousands of random combinations, evaluates the UCB score for each, and then sorts and takes the top. The UCB score is calculated independently for each sample and therefore this approach will have the tendency to converge to similar solutions with high UCB scores. This results in low diversity experiments that provide limited new information.

There is a vast amount of literature on this batch UCB setting, e.g.:

<https://proceedings.mlr.press/v70/daxberger17a.html> . Incorporating batch UCB approaches would likely improve the METIS method because each design-test-learn cycle would generate richer data for subsequent rounds.

Response:

We thank the reviewer for the suggestion regarding batch upper confidence bound. We tested the Batch Upper Confidence Bound algorithm on a test function and saw a slightly better performance compared to our UCB score. We therefore implemented this algorithm as an optional feature of METIS (to be defined by users instead of UCB). The reason we implemented it as an optional feature is that Batch UCB can be costly in term of time.

Action:

We added the following to the Methods section of the manuscript and created Supplementary Fig. 22 to show the results:

P16L38-40: “See Supplementary Fig. 22 for an optional scoring (can be defined when using METIS), Batch UCB that can generate richer combinations for subsequent rounds.”

Supplementary Fig. 22: Batch Upper Confidence Bound (UCB) for scoring objective functions. We chose the Batch Upper Confidence Bound¹⁴ algorithm because of its high compatibility and ease of implementation to the current version of METIS. This algorithm can be summarized as following to obtain a batch size of k : In each round the top UCB score sample is chosen, added to the training dataset with its predicted value, this process is repeated for k time. This algorithm avoids points similar to what already chosen in a batch since they are going to have low standard variation parameters in UCB score. We evaluated the performance of the Batch UCB algorithm **(b)** compared to UCB algorithm **(a)** on a test function (Branin Function¹⁵). Three independent simulations were repeated and plot for **(a)** and **(b)**. The Batch UCB algorithm shows a performance slightly better than UCB and since it can be costly in term of time, we added it as an optional setting to METIS.